# Interplay of cell–cell contacts and RhoA/MRTF-A signaling regulates cardiomyocyte identity

Tatjana Dorn[1,†], Jessica Kornherr[1,†], Elvira I Parrotta[1,2,†] (ID), Dorota Zawada[1], Harold Ayetey[3,4] (ID), Gianluca Santamaria[2] (ID), Laura Iop[1], Elisa Mastantuono[5], Daniel Sinnecker[1,6], Alexander Goedel[1,6], Ralf J Dirschinger[1], Ilaria My[1], Svenja Laue[1], Tarik Bozoglu[1], Christian Baarlink[7], Tilman Ziegler[1], Elisabeth Graf[5], Rabea Hinkel[1,6,8], Giovanni Cuda[2], Stefan Kääb[9], Andrew A Grace[4,10], Robert Grosse[7], Christian Kupatt[1,6] (ID), Thomas Meitinger[5], Austin G Smith[3,10] (ID), Karl-Ludwig Laugwitz[1,6,*] (ID) & Alessandra Moretti[1,6,**] (ID)

## Abstract

Cell–cell and cell–matrix interactions guide organ development and homeostasis by controlling lineage specification and maintenance, but the underlying molecular principles are largely unknown. Here, we show that in human developing cardiomyocytes cell–cell contacts at the intercalated disk connect to remodeling of the actin cytoskeleton by regulating the RhoA-ROCK signaling to maintain an active MRTF/SRF transcriptional program essential for cardiomyocyte identity. Genetic perturbation of this mechanosensory pathway activates an ectopic fat gene program during cardiomyocyte differentiation, which ultimately primes the cells to switch to the brown/beige adipocyte lineage in response to adipogenesis-inducing signals. We also demonstrate by *in vivo* fate mapping and clonal analysis of cardiac progenitors that cardiac fat and a subset of cardiac muscle arise from a common precursor expressing Isl1 and Wt1 during heart development, suggesting related mechanisms of determination between the two lineages.

**Keywords** cardiac fat; cardiac progenitors; lineage conversion; MRTF/SRF; RhoA/ROCK signaling
**Subject Categories** Cell Adhesion, Polarity & Cytoskeleton; Development & Differentiation; Stem Cells
**The EMBO Journal (2018) 37: e98133**

## Introduction

Developmental commitment of cells and tissues is governed by combinations of lineage-specific transcription factors that act as master switch genes defining and reinforcing cell type-specific gene expression patterns. It has been demonstrated that artificial ectopic expression of lineage-specific master genes is sufficient to reprogram one cell type into another, both *in vitro* and *in vivo* (Takahashi *et al*, 2007; Pang *et al*, 2011; Ahfeldt *et al*, 2012; Qian *et al*, 2012). In some vertebrates and invertebrates, conversion between cell types takes place naturally during developmental and regeneration processes in response to local signals (Sisakhtnezhad & Matin, 2012). In humans or mammals, only limited evidence suggests that spontaneous cell fate changes could occur under pathological conditions (Park *et al*, 2006; El Agha *et al*, 2017; Plikus *et al*, 2017). Learning the molecular principles of natural cell lineage conversion will be critical for understanding how to reset cellular identities for application in cell therapy and regenerative medicine.

Direct transdifferentiation of cardiac muscle cells into adipocytes has been postulated in arrhythmogenic right ventricular cardiomyopathy (ARVC; d'Amati *et al*, 2000), a heritable cardiac disease pathologically characterized by fibroadipocytic replacement of cardiac myocytes (CMs) leading to right ventricular failure, arrhythmias, and sudden cardiac death (Basso *et al*, 2009). Mutations in genes encoding proteins of the desmosome and the *area composita* are collectively responsible for ~50% of ARVC cases (Delmar & McKenna, 2010). Elegant lineage-tracing studies in mice harboring

1   Klinik und Poliklinik Innere Medizin I, Klinikum rechts der Isar – Technical University of Munich, Munich, Germany
2   Department of Experimental and Clinical Medicine, Medical School, University of Magna Grecia, Catanzaro, Italy
3   Wellcome Trust – Medical Research Council Stem Cell Institute, University of Cambridge, Cambridge, UK
4   Papworth Hospital NHS Foundation Trust, Cambridge, UK
5   Institute of Human Genetics, Klinikum rechts der Isar – Technical University of Munich, Munich, Germany
6   DZHK (German Centre for Cardiovascular Research) - partner site Munich Heart Alliance, Munich, Germany
7   Pharmacology Institute, Philipps University Marburg, Marburg, Germany
8   IPEK Institute for Cardiovascular Prevention, Klinikum der Universität München – Ludwig-Maximillians-Universität, Munich, Germany
9   Medizinische Klinik und Poliklinik I, Klinikum der Universität München – Ludwig-Maximillians-Universität, Munich, Germany
10  Department of Biochemistry, University of Cambridge, Cambridge, UK
    *Corresponding author. Tel: +49 89 4140 2350; E-mail: laugwitz@mytum.de
    **Corresponding author. Tel: +49 89 4140 6907; E-mail: amoretti@mytum.de
    †These authors contributed equally to this work

conditional ablation of the desmosomal protein desmoplakin in adult CMs or embryonic cardiac progenitors have provided first evidence that pathological fat in ARVC can arise from mature CMs, and this process is "primed" early during organogenesis (Lombardi *et al*, 2009). As the heart is constantly exposed to a high level of mechanical strain from the beginning of its development, this suggests that coordination of cell adhesion is likely a key process whereby mechanics could regulate cellular identity of CMs. Indeed, cell–cell and cell–matrix contacts are known to integrate with cytoskeletal organization to influence response to mechanical cues and regulate developmental fate and lineage specification. However, whether similar principles also operate during cell lineage conversion remains elusive.

Here, we used CMs from ARVC patients harboring cell–cell adhesion defects to evaluate the role of mechanical sensing in controlling cardiac muscle identity and elucidate how force-bearing cell–cell contacts link to changes of transcriptional programs during pathological myocyte-to-adipocyte switch. Furthermore, using *in vivo* fate mapping and clonal analysis of cardiac progenitors we show that cardiac fat and a subset of CMs arise from a common Isl1/Wt1-expressing precursor during development.

# Results

## CMs carrying pathological mutations in desmosomal proteins convert into brown/beige adipocytes *in vitro*

We generated induced pluripotent stem cells (iPSCs) from an ARVC patient who carried a heterozygous frameshift mutation (c.1760delT; p.V587Afs*655) in the *PKP2* gene encoding the desmosomal protein plakophillin (Appendix Fig S1). We coaxed iPSCs to differentiate into CMs with a ~97% purity and analyzed the ability of CMs to convert into adipocytes when cultured under conditions mimicking the mechanical strain of the heart (50 kPa substrate stiffness and 1 Hz electrical pacing) and exposed to a lipogenic milieu favoring the fatty acid oxidation-based metabolism found in both adult CMs and adipocytes (see Materials and Methods).

Mutated CMs had normal levels of *PKP2* transcript (Fig EV1A), but plakophillin protein was reduced by up to 50% compared to wild-type (wt) control cells, with no C-terminal truncated form detectable (Fig EV1B). Consistently, immunohistochemistry revealed decreased PKP2 expression at the plasmamembrane, concurring with a thin and interrupted desmosome structure (Fig 1A). Such alterations at the intercalated disks were further confirmed by immunodetection of desmoplakin (Fig EV1C), indicating defects of mutant CMs in establishing cell–cell junctions. Immunofluorescence analysis of cardiac troponin T (cTNT), a protein marking CM sarcomeres, in conjunction with the lipid stain Oil Red O (ORO) revealed gradual morphological and structural changes in diseased CMs over time in culture (Fig 1B). While wt cells showed a stable myocytic phenotype over a 4-week period with only little lipid accumulation indicative of lipogenesis, a progressive disarray of myofilaments and development of enlarged multilocular lipid droplets were detected in mutated CMs (Fig 1B–D), suggesting an ongoing loss of myocytic identity and acquisition of fat cell phenotype. Lipid-filled adipocyte-like cells lacking sarcomeres and morphologically resembling brown/beige adipocytes (multilocular

lipid droplet morphology) were observed in mutated cells from day 21, with their number increasing over time (Fig 1B and E).

Quantitative RT–PCR for a panel of CM, preadipocyte, and adipocyte markers confirmed that the morphological transformation of *PKP2* mutant CMs indeed was accompanied by concurrent changes in cell type-specific gene expression (Fig 1F). Interestingly, when compared to wt CMs, mutated cells at baseline (i.e., before initiation of pacing and culture in lipogenic milieu) already expressed higher levels of key genes directing preadipocyte differentiation, such as *PPARγ*, *C/EBPα* (Farmer, 2006), and *PRDM16*, a transcriptional regulator of *PPARγ* that specifies the brown fat lineage (Seale *et al*, 2008). Moreover, *TCF21*, a transcription factor that is expressed in preadipocytes and has been shown to suppress myogenesis (Timmons *et al*, 2007), was also upregulated in mutated CMs, suggesting that mutants might be less faithfully restricted to the CM phenotype and already primed for the adipocyte lineage. In mutant cultures, genes of terminally differentiated adipocytes, such as *FABP4* and *ADIPOQ* (Gesta *et al*, 2007), became progressively upregulated over time, while CM genes (*TNNT2*, *MYH7,* and *ACTC1*) were gradually downregulated (Fig 1F). Genes associated with apoptosis (*BAX1*, *BIM,* and *FAS*) remained unchanged (Fig EV1D), suggesting that the morphological/structural and transcriptional changes seen in the pathological settings were not consequences of apoptotic death and selection but rather the result of cell type conversion. Intriguingly, in addition to *PRDM16*, other brown/beige-fat-cell-specific genes, including *UCP1*, *CIDEA,* and *ELOVL3* (Cohen & Spiegelman, 2015; Ikeda *et al*, 2018), showed elevated expression after prolonged culture of mutant CMs, whereas genes enriched in white fat (*FBN1* and *NRIP1*; Gesta *et al*, 2007) were not upregulated (Fig EV1D). Remarkably, expression levels of brown/beige adipocyte markers in mutated cells were comparable to those measured in human iPSC-derived brown/beige adipocytes (Ahfeldt *et al*, 2012; Guenantin *et al*, 2017; Fig EV1E), suggesting a "true" conversion toward the brown/beige fat lineage.

To address the stability of the myocyte-to-adipocyte switch in pathological conditions, we cultured wt and mutated cells for up to 4 additional weeks and evaluated the expression of PPARγ and cTNT as well as the exhibition of a mature functional brown/beige adipocyte phenotype. Immunofluorescence analysis for PPARγ, a master regulator of adipogenesis, in conjunction with lipid BODIPY stain at 8-week culture revealed that roughly 40–50% of mutant cells maintained the morphological appearance of mature brown/beige adipocytes, expressed high levels of nuclear PPARγ, and were negative for cTNT (Fig 1G). In contrast, wt cells lacked PPARγ and kept the typical myocytic phenotype with well-organized cTNT⁺ sarcomeres and regular contractions (Fig 1G). Furthermore, the ability to break down triglycerides and release glycerol was significantly higher in mutant cells compared to control and further increased upon forskolin treatment (Fig 1H), indicative of functional adipocytes. As characteristic feature, brown/beige adipocytes carry out efficient thermogenesis under β-adrenergic stimulation (Cereijo *et al*, 2015). Consistently, upregulation of thermogenic genes upon treatment with 8-Br-cAMP was similar between mutant cells and iPSC-derived brown/beige adipocytes (Fig EV1F), confirming a brown/beige fat identity of the converted cells.

Taken together, these results indicate that a direct conversion of CMs into adipocytes with brown/beige fat identity can take place in the presence of pathological mutations that weaken cell–cell

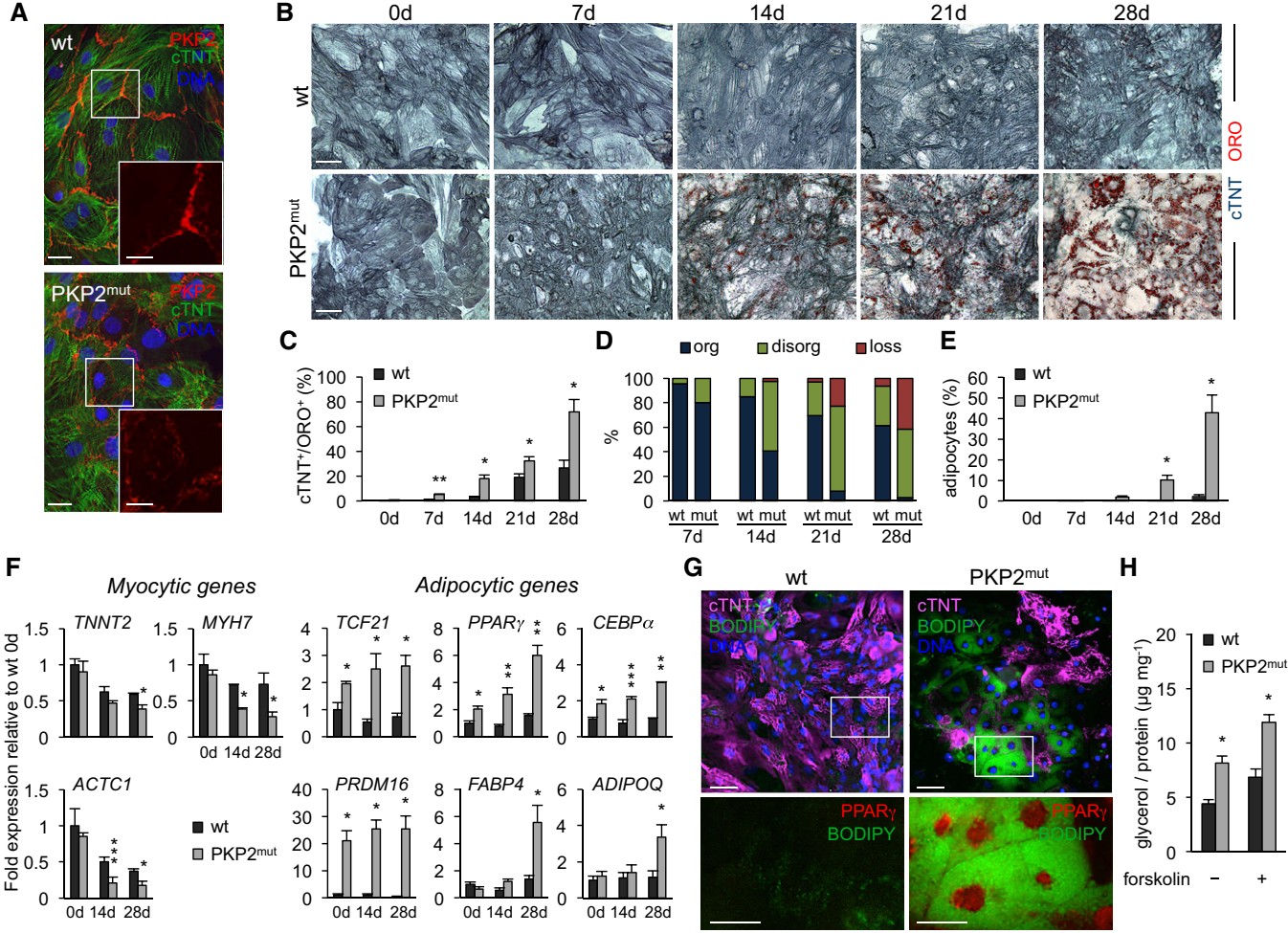

**Figure 1.  CMs with pathological *PKP2* mutation convert into adipocytes *in vitro*.**

A    Immunostaining of cTNT (green) and PKP2 (red) in wt and PKP2^mut CMs. Scale bars, 25 μm. Insets show a magnified image of PKP2 expression; scale bars, 10 μm.

B    Immunodetection of cTNT (blue) and lipid stain ORO (red) in wt and PKP2^mut CMs over time in culture. Scale bars, 50 μm.

C–E    Bar graphs show the percentage of cTNT^+/ORO^+ cells (C), cTNT^+/ORO^+ cells with organized (org), disorganized (disorg) and dissolving sarcomeres (loss) (D), and adipocyte-like cells (E) in wt and PKP2^mut CMs over time (n = 3; N = 45–300 cells per time point in each group; *P < 0.05, **P < 0.001 vs. wt at the same time point; t-test).

F    qRT–PCR analysis of myocytic and adipocytic genes in wt and PKP2^mut CMs over time in culture (n = 3; *P < 0.05, **P < 0.01, ***P < 0.001 vs. wt at the same time point, t-test).

G    Immunodetection of cTNT (magenta) and lipid stain BODIPY (green) in wt and PKP2^mut CMs after 8 weeks of culture. Scale bars, 50 μm. Lower panels show magnified images of areas indicated by the white box for BODIPY (green) and PPARγ (red). Scale bars, 25 μm.

H    Glycerol release of wt and PKP2^mut CMs after 8 weeks in culture in the absence or presence of 10 μM forskolin (n = 3; *P < 0.01 vs. wt; t-test).

Data information: All data are shown as means ± SEM.
Source data are available online for this figure.

adhesion. Considering that the efficiency of the conversion process is only ~40%, we reasoned that a certain subpopulation of CMs might be "primed" for remodeling of transcriptional networks depending on pre-existing transcriptional or epigenetic status likely acquired during the cell fate specification processes throughout organ development.

## Cardiac fat and a subpopulation of CMs share common embryonic Isl1/Wt1-expressing progenitors

To address directly whether a common progenitor exists between a subset of CMs and fat cells, we performed lineage-tracing

experiments in mice. Human pathological cardiac adipogenesis in ARVC occurs mainly in the right ventricle (RV), which arises from a specific population of cardiac progenitors—known as second heart field (SHF) progenitors—marked by the transcription factor Isl1. Thus, we crossed mice that express the Cre recombinase under control of the *Isl1* locus (*Isl1^{Cre/+}*; Yang *et al*, 2006) with double-fluorescent Cre reporter mice, in which transgene expression of membrane-targeted tdTomato (mT) from the *Rosa26* locus converts to expression of membrane-targeted green fluorescent protein (mG) in a Cre-dependent manner (*R26^{mTmG/+}*), allowing tracing of *Isl1*-expressing descendants by mG expression. Highly efficient mG labeling was observed in RV CMs and in the vast majority of

perilipin (Plin1)-positive cells of cardiac fat depots—which are known to have brown character and are limited to a very specific location between the atrial and ventricular chambers on the dorso-lateral sides of the heart (called AV groove; Yamaguchi *et al*, 2015; Fig 2A). AV groove fat originates from the proepicardial organ (PEO), a transient extra-cardiac cluster of progenitors that arises between embryonic day (E) 9 and 10.5 at the ventro-caudal base of the developing mouse heart (Liu *et al*, 2014; Yamaguchi *et al*, 2015). PEO contributes cells to various cardiac lineages and is marked by expression of multiple genes, including the transcription factor Wilm's tumor 1 (Wt1; Brade *et al*, 2013; Risebro *et al*, 2015). Immunofluorescence analysis for Isl1 and Wt1 in the PEO of $Isl1^{Cre/+};R26^{mTmG/+}$ embryos at E9.5 revealed expression of the lineage marker mG in almost half of the Wt1$^+$ cells (44 ± 5%, $n = 5$ embryos) and complete absence of Isl1 expression throughout the tissue (Fig 2B), suggesting that a subset of Wt1$^+$ proepicardial cells derive from Isl1$^+$ progenitors that have turned off Isl1 at the PEO state. Indeed, co-staining of Wt1 and Isl1 proteins at developmental stages preceding PEO formation identified a rare population of double Wt1/Isl1-expressing cells in the caudal lateral mesoderm at E8 (50–70 cells/embryo, Fig 2C). We traced the descendants of these cells using inducible $Isl1^{MerCreMer/+}$ and $Wt1^{CreERT2/+}$ mice, in which Cre recombinase requires tamoxifen to be active (Feil *et al*, 1997; Laugwitz *et al*, 2005). We performed maternal injection of low doses of tamoxifen at E7.5 and analyzed E9.5 embryos and 4-week postnatal hearts. At E9.5, we found an overlap of mG$^+$ cells between the two mouse lines in the epicardium, a PEO derivative, while exclusive marking of SHF progenitors/cardiac myocytes and PEO progenitors was observed in the $Isl1^{MerCreMer/+};R26^{mTmG/+}$ and $Wt1^{CreERT2/+};R26^{mTmG/+}$ lines, respectively (Fig EV2). Examination of adult hearts demonstrated mG labeling of both AV groove adipocytes and CMs in 22% of $Isl1^{MerCreMer/+};R26^{mTmG/+}$ and 86% of $Wt1^{CreERT2/+};R26^{mTmG/+}$ animals analyzed (Figs 2D and EV3). Furthermore, other cardiac lineages known to originate from PEO progenitors expressed mG in $Wt1^{CreERT2/+};R26^{mTmG/+}$ adult mice (Appendix Fig S2A), confirming that tamoxifen injection at E7.5 labels true Wt1$^+$ proepicardial precursors prior to the development of the PEO. No activation of Cre was observed in absence of tamoxifen (Appendix Figs S2B and S3).

To assess the differentiation potential of early double Isl1$^+$/Wt1$^+$ progenitors into the myocytic and adipocytic lineages at the clonal level, we took advantage of mouse embryonic stem cells (ESCs) and of a cardiac mesenchyme culture (CMC) system that we previously established to efficiently promote the self-renewal of early Isl1$^+$ cardiac precursors (Laugwitz *et al*, 2005; Moretti *et al*, 2006; Qyang *et al*, 2007). We derived ESCs from the blastocysts of $Isl1^{Cre/+}$ mice crossed with the $R26^{YFP/+}$ indicator mice that express yellow fluorescent protein (YFP) from the *Rosa26* gene locus upon Cre-mediated recombination (Fig 3A). Within 4–5 days of embryoid body (EB) differentiation, the first Isl1$^+$ progenitors arose, as demonstrated by YFP expression and detection of *Isl1* transcript (Fig 3B and C). This coincided with the upregulation of other key markers of early cardiac progenitors, such as *Nkx2-5* and *Gata4* (Moretti *et al*, 2006; Wu *et al*, 2006; Fig 3C). Fluorescence-activated cell sorting (FACS) of YFP$^+$ cells from 4d EBs, followed by exposure to pro-myogenic or pro-adipogenic culture conditions, demonstrated the differentiation potential of this population into CMs and adipocytes (Fig 3D and E). We

then clonally expanded single FACS-sorted Isl1/YFP-expressing precursors on CMC feeder layers and analyzed their bi-differentiation potential and gene expression profile (Fig 3A). Immunohisto-chemistry analysis after 6 days of co-culture revealed that ~50% of YFP$^+$ clones co-expressed Isl1 and Wt1 (Fig 3F). Interestingly, when induced to differentiate, only clones that expressed both *Isl1* and *Wt1* gave rise to the two lineages, as demonstrated by the appearance of striated muscle cells expressing cTNT and lipid-filled adipocytes expressing PPARγ (Fig 3G and H). Differentiation into one lineage was observed only for CMs, triggered in clones that were exclusively positive for *Isl1* and *Nkx2-5*, a key marker of myocytic fate (Moretti *et al*, 2006; Wu *et al*, 2006; Fig 3H). Moreover, further treatment of differentiating CM cultures with pro-adipogenic medium induced significant accumulation of lipid droplets merely in CMs from Isl1$^+$/Wt1$^+$ clones (Appendix Fig S4), indicating a higher plasticity of these CMs toward the adipocytic fate.

Taken together, these results suggest that the molecular signature of Isl1$^+$/Wt1$^+$ defines a cardiac "myo-adipo progenitor cell" *in vitro*, which is distinct from Isl1$^+$/Nkx2-5$^+$ myocytic precursors. Furthermore, they hint to an important role of Wt1 in programming Isl1$^+$ cardiac precursors into the adipogenic lineage and a possible involvement in "priming" CMs to spontaneously adopt a brown/beige adipocyte phenotype in specific disease-associated conditions.

### WT1 is activated during pathological myocyte-to-adipocyte conversion but is not sufficient to induce PPARγ, the molecular trigger of the adipogenic phenotype

To investigate the potential contribution of WT1 to the pathological myocyte-to-adipocyte lineage conversion observed in *PKP2* mutant CMs, we first analyzed its expression and cellular distribution before and during induction of the adipocyte switch (Fig 4A). WT1 is predominantly a nuclear protein in most expressing tissues and functions both as a transcriptional activator/co-activator or repressor of gene expression (Hohenstein & Hastie, 2006); however, owing to its ability to bind RNA (Hohenstein & Hastie, 2006) and actin (Dudnakova *et al*, 2010), it can also be detected in the cytoplasm of many cells, mainly localized on cytoskeleton-bound polysomes (Dudnakova *et al*, 2010). Interestingly, differently from mouse CMs that lack expression of Wt1 at both embryonic and adult stages (Zhou *et al*, 2008), a strong WT1 cytoplasmic immunostaining signal has been reported in developing CMs of human fetuses (Ambu *et al*, 2015).

Immunocytochemistry in human iPSC-derived CMs demonstrated expression of WT1 in both wt and *PKP2* mutated cells, which was predominantly localized in the cytoplasm in a filament-like pattern (Figs 4A, and EV4A and B). Remarkably, WT1 transcript and protein levels were significantly higher in *PKP2* mutant CMs (Figs EV4A and B, and 4A). Moreover, lipogenic culture conditions induced nuclear translocation of WT1 exclusively in mutated cells, which ultimately coincided with marked myofibril disarray and PPARγ activation (Fig 4A). In line with these observations, immunohistological inspection of WT1 expression in adult human myocardium from ARVC patients revealed a nuclear localization of WT1 protein in some of the CMs adjacent to the fibrofatty tissue, while only cytosolic-localized protein could be detected in heart

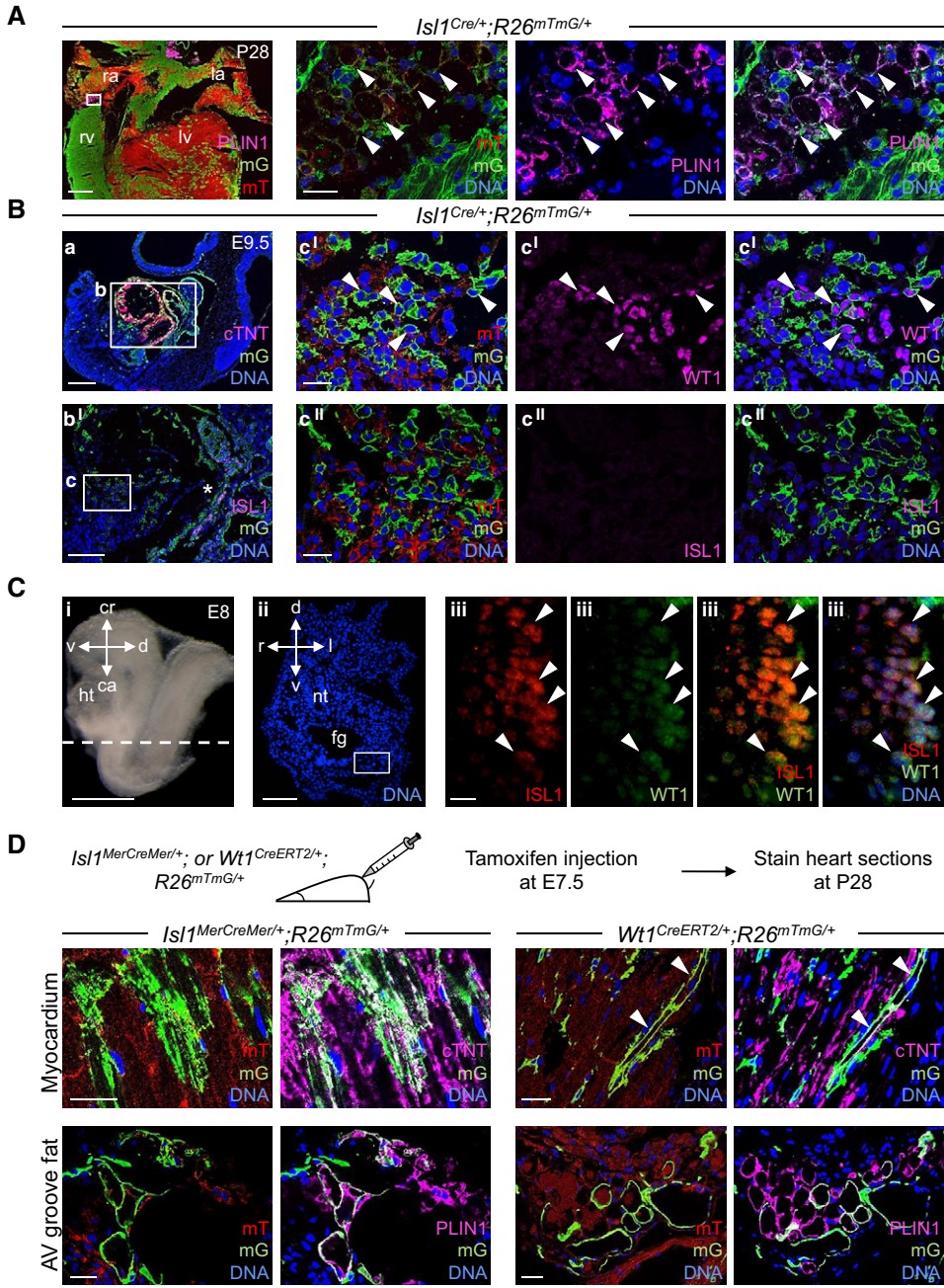

**Figure 2.  Cardiac muscle and fat derive from Isl1/Wt1-expressing progenitors.**

A   Immunofluorescence analysis of Isl1 derivatives in *Isl1^Cre/+^;R26^mTmG/+^* mouse hearts at postnatal d28 (P28): Plin1 (magenta), membrane GFP (mG, green), and membrane Tomato (mT, red). The three panels on the right show a magnified view of the area indicated by the white box in the left panel. Arrows indicate Plin1⁺/mG⁺ adipocytes. la, left atrium; lv, left ventricle; ra, right atrium; rv, right ventricle. Scale bars, 500 μm (left panel), 25 μm (right panels); *n* = 3.

B   Immunostaining of cTNT, Isl1 or Wt1 (all magenta), mG (green), and mT (red) in sagittal, sequential sections of *Isl1^Cre/+^;R26^mTmG/+^* embryos at E9.5. The boxed region in panel a is shown at higher magnification in the consecutive section in b¹. c¹ and c¹¹ are magnifications of the boxed area in b¹ (in consecutive sections). Isl1-Cre-mediated mG labeling was observed in CMs of the forming heart (a, box b), cardiac progenitors (b¹, marked by the asterisk), and a proportion of Wt1⁺ proepicardial cells in the PEO (c¹, indicated by arrows, and c¹¹). Scale bars, 250 μm (a), 150 μm (b¹), 25 μm (c¹ and c¹¹); *n* = 3.

C   Immunostaining for Isl1 and Wt1 in transverse sections of E8 embryos. A representative section corresponding to the position indicated by the dashed line drawn through the adjacent embryo view (i) is shown in panel ii after Hoechst 33258 staining of nuclei (blue). Panels iii show the boxed area in panel ii after Isl1 (red) and Wt1 (green) co-immunostaining. Isl1⁺/Wt1⁺ progenitors and marked by the arrows in (iii). Scale bars, 2 mm (i), 200 μm (ii), 10 μm (iii). ca, caudal; cr, cranial; d, dorsal; fg, foregut; ht, heart; l, left; nt, neural tube; r, right; v, ventral; *n* = 3.

D   Temporal restriction of Cre-mediated labeling of Isl1⁺ and Wt1⁺ progenitors and their derivatives using tamoxifen-inducible *Isl1^MerCreMer/+^;R26^mTmG/+^* and *Wt1^CreERT2/+^;R26^mTmG/+^* mouse lines. Labeling was induced by tamoxifen treatment at E7.5, and heart sections were analyzed at postnatal day 28 (P28). Shown are representative immunostainings of cTNT or PLIN1 (magenta), mG (green), and mT (red) in sections of *Isl1^Cre/+^;R26^mTmG/+^* (left panels) and *Wt1^CreERT2/+^;R26^mTmG/+^* (right panels). Arrows indicate cTNT⁺/mG⁺ CMs. Scale bars, 25 μm; *n* = 3 per genotype.

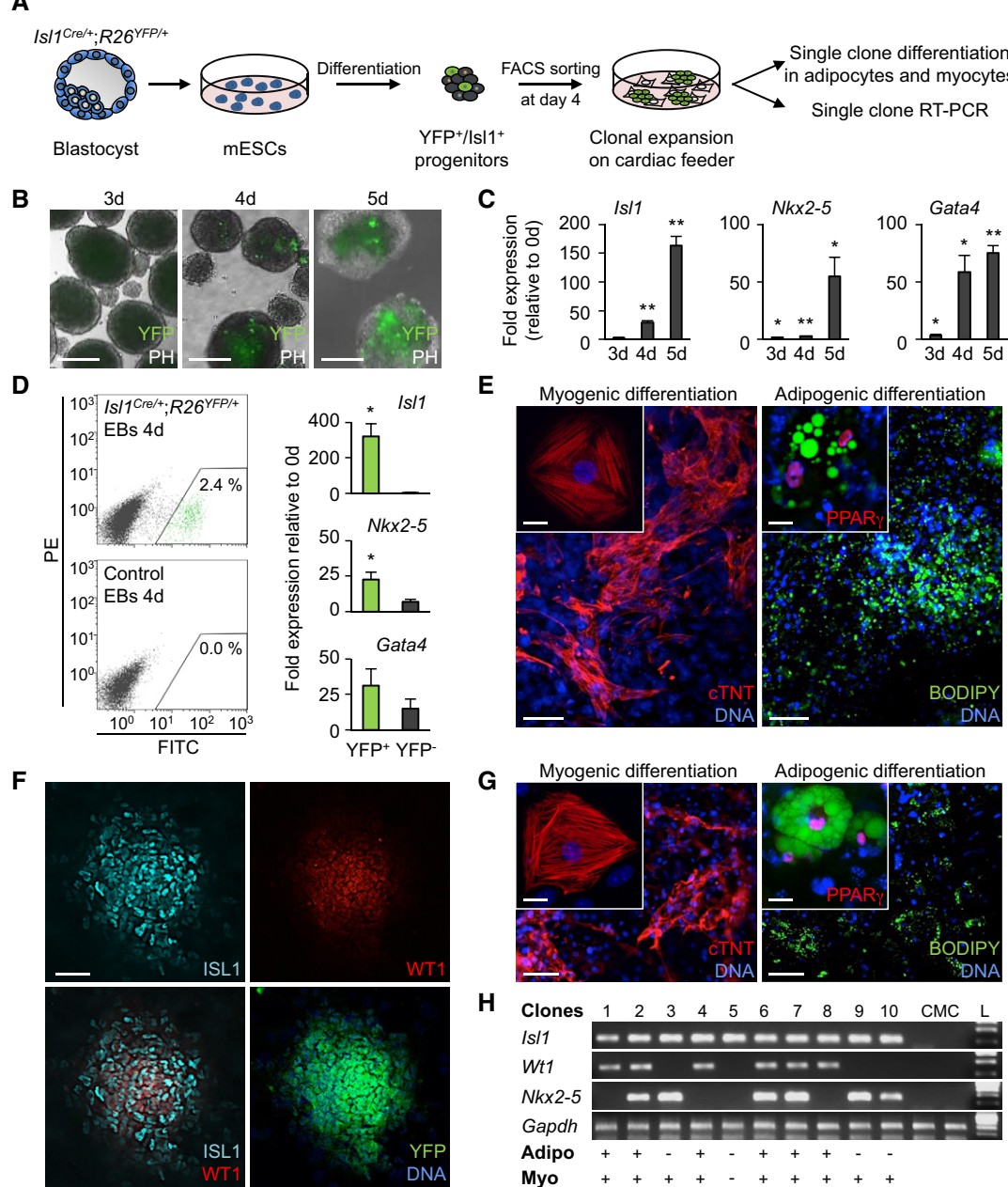

**Figure 3. A single Isl1/Wt1-expressing progenitor gives rise to both CMs and adipocyte lineages *in vitro*.**

A    Scheme for isolation of Isl1+/YFP+ cardiac precursors from the blastocyst-derived mouse *Isl1Cre/+;R26YFP/+* ESCs and their clonal analysis.

B, C    Assessment of YFP expression by bright field/epifluorescence microscopy (B) and qRT–PCR analysis of cardiac progenitor markers (C) during EB differentiation of *Isl1Cre/+;R26YFP/+* ESCs. Scale bars, 200 μm; *n* = 4; *P < 0.05, **P < 0.01 vs. d0; *t*-test. Data are shown as means ± SEM.

D    Flow cytometry analysis of YFP expression and qRT–PCR analysis of cardiac progenitor markers in YFP+ and YFP− cells sorted from 4-day-old *Isl1Cre/+;R26YFP/+* EBs. FITC, fluorescein isothiocyanate; PE, phycoerythrin. *n* = 3; *P < 0.05 vs. YFP−; *t*-test. Data are shown as means ± SEM.

E    Immunostaining of cTNT (red) and PPARγ (red) in conjunction with lipid stain BODIPY (green) of sorted cells from 4-day-old *Isl1Cre/+;R26YFP/+* EBs after differentiation in pro-myogenic and pro-adipogenic culture conditions. Scale bars, 50 μm. The insets show higher magnifications, scale bars, 25 μm.

F    Immunostaining of Isl1 (cyan), Wt1 (red), and YFP (green) in a representative clone derived from a single Isl1+ progenitor sorted from 4-day-old *Isl1Cre/+;R26YFP/+* EBs and expanded on CMCs for 6 days. Scale bar, 100 μm.

G    Immunostaining of cTNT (red) and PPARγ (red) in conjunction with BODIPY (green) in cell derivatives from a single Isl1+/Wt1+ clone after differentiation in pro-myogenic (left) or pro-adipogenic (right) culture conditions for 6 and 28 days, respectively. Scale bars, 50 μm. The inlays show higher magnifications; scale bars, 25 μm.

H    RT–PCR profile of 10 representative clones after expansion on CMCs for 6 days. Isl1+/Wt1+ clones differentiated into both adipocytic and myocytic lineages, and Isl1+/Nkx2-5+/Wt1− clones differentiated only into the myocytic lineage. CMC cDNA was used as a negative control. Adipo, adipocytes; CMC, cardiac feeder cells; L, DNA ladder; Myo, myocytes.

Source data are available online for this figure.

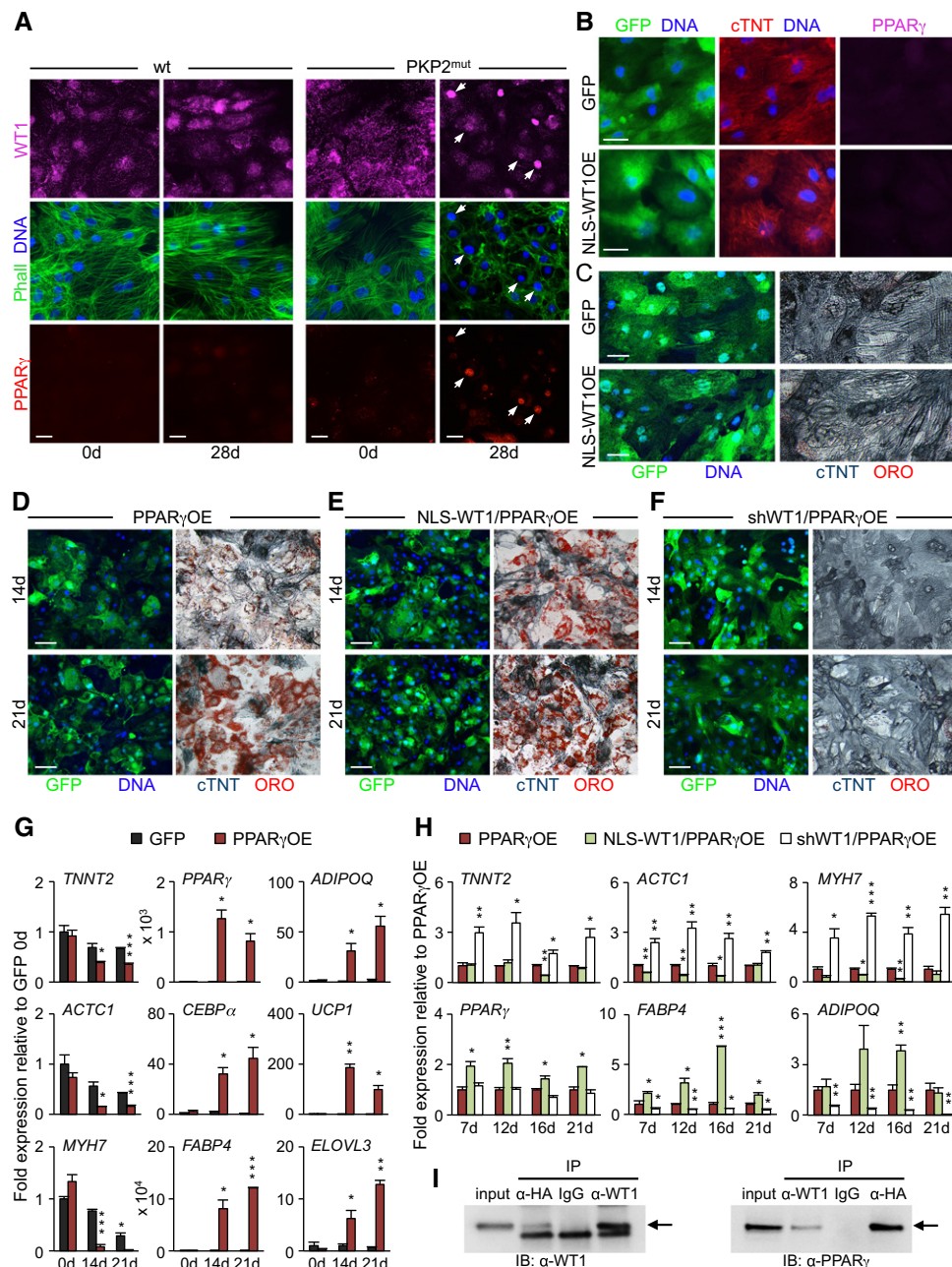

**Figure 4.  Cooperative role of WT1 and PPARγ in cardiomyocyte-to-adipocyte conversion.**

A     Immunostaining of endogenous WT1 (magenta), F-actin (Phalloidin, green), and PPARγ (red) in wt and PKP2$^{mut}$ CMs at 0 and 28 days in culture. Arrows indicate PKP2$^{mut}$ cells with nuclear translocation of WT1, sarcomere disarray, and PPARγ activation. Scale bars, 25 μm.

B, C   Immunocytochemistry of wt CMs after 28-day infection with lentiviral constructs encoding the nuclear-tagged form of WT1 (NLS-WT1) plus GFP or GFP alone showing GFP (green), cTNT (red) and PPARγ (magenta) (B), and GFP (green, left) and cTNT (blue) in conjunction with lipid stain ORO (red, right) (C). Scale bars, 25 μm (B) and 50 μm (C).

D–F   Immunocytochemistry of wt CMs after lentiviral overexpression of PPARγ alone (PPARγOE, D), PPARγ and NLS-WT1 (NLS-WT1/PPARγOE, E) or PPARγ, and shRNA targeting WT1 (shWT1/PPARγOE, F) at 14 (top) and 21 (bottom) days after infection. All constructs co-express GFP. Representative images show GFP (green, left) and cTNT (blue) in conjunction with ORO stain (red, right) in the same cells. Scale bars, 50 μm.

G, H   qRT–PCR analysis of myocytic and adipocytic genes in PPARγOE (G), NLS-WT1/PPARγOE (H), and shWT1/PPARγOE (H) conditions at the indicated time points; *n* = 3 (PPARγOE, NLS-WT1/PPARγOE) and *n* = 4 (shWT1/PPARγOE). *$P < 0.05$, **$P < 0.01$, ***$P < 0.001$ vs. GFP control (G) or PPARγOE (H) at the same time point (*t*-test). Data are shown as means ± SEM.

I     Wt CMs were transduced with HA-tagged PPARγ and NLS-WT1 and cultured in adipogenic medium for 14 days. Nuclear cell lysates were precipitated using anti-HA or anti-WT1 followed by immunoblot (IB) analysis with an anti-WT1 or anti-PPARγ, and IgG was used as control; *n* = 3. Arrows indicate WT1 (left panel) and PPARγ (right panel) bands.

Source data are available online for this figure.

tissue samples from individuals lacking mutations in cell–cell junctional proteins (Fig EV4C and D).

We next explored whether nuclear shuttling of WT1 would be sufficient to induce PPARγ expression and whether PPARγ functions as myocyte-to-adipocyte-specific master switch. We transduced wt CMs with lentiviral constructs encoding either a nuclear-tagged form of WT1 (NLS-WT1) or PPARγ together with GFP or GFP alone (as control; Appendix Fig S5A and B) and analyzed the morphological and molecular dynamic changes occurring during culture of the cells in lipogenic milieu. We obtained a transduction efficiency of ~80%, as measured by the percentage of GFP$^+$ cells expressing nuclear WT1 or PPARγ (Appendix Fig S5A and B). Upon overexpression of NLS-WT1, CMs maintained a stable myocytic phenotype with well-organized sarcomeres (Fig 4B and C). No activation of PPARγ or other adipogenic-specific genes (Fig 4B and Appendix Fig S5C) nor appearance of lipid droplets was observed even after prolonged culture (Fig 4C), suggesting that nuclear localization of WT1 is not sufficient to induce myocyte-to-adipocyte conversion in wt settings. However, forced expression of PPARγ resulted in myofibril disarrangement and C/EBPα activation in all infected CMs by 7 days (Appendix Fig S5D). Within 3 weeks, the vast majority of PPARγ-infected CMs had lost cTNT expression and acquired a lipid-filled phenotype resembling brown/beige adipocytes (Fig 4D). Gene expression analysis demonstrated a progressive decrease in myogenic transcripts in PPARγ-overexpressing cells over time, which concurred with a strong upregulation of terminally differentiated adipocyte markers, including FABP4 and ADIPOQ at day 21 (Fig 4G). PPARγ overexpression also induced robust activation of genes specific for brown/beige fat, such as UCP1 and ELOVL3 (Fig 4G), suggesting a brown/beige fat identity of the switched cells. Intriguingly, when NLS-WT1 was co-overexpressed with PPARγ—conditions more closely resembling the situation seen in PKP2 mutant CMs—we observed an acceleration of the myocyte-to-adipocyte conversion process, as indicated by the faster appearance of lipid-filled adipocyte-like cells (Fig 4E). At the mRNA level, we measured a higher upregulation of FABP4 and ADIPOQ in NLS-WT1/PPARγ-overexpressing cells compared to PPARγ overexpression alone throughout the 3-week culture period (Fig 4H). Interestingly, PPARγ expression was likewise significantly elevated in the NLS-WT1/PPARγ condition (Fig 4H), indicating a possible PPARγ-dependent regulation of endogenous PPARγ by WT1.

To further investigate the potential cooperative role of WT1 and PPARγ in the cardiomyocyte-to-adipocyte switch, we first tested whether knockdown of endogenous WT1 in PPARγ-overexpressing wt CMs would negatively affect their conversion. Indeed, by day 21, cultures expressing shRNA targeting WT1 showed a striking reduction in lipid-filled adipocyte-like cells and levels of adipocytic genes (Fig 4F and H), while scrambled shRNA had no effects (Appendix Fig S6). Co-immunoprecipitation analysis in NLS-WT1/PPARγ-overexpressing wt CMs ultimately demonstrated a physical interaction of the two proteins in the nucleus of converting cells (Fig 4I). Finally, we evaluated the effects of NLS-WT1 and PPARγ overexpression in adult mouse CMs that lack endogenous Wt1. Using adeno-associated virus-mediated delivery of the transgenes in $Myh6^{Cre/+}$;$R26^{mTmG/+}$ mice, we observed emergence of intramyocardial lipid-filled/Plin$^+$ adipocyte-like cells labeled by mG exclusively in hearts of animals that received both viruses (Fig EV5).

Altogether, these findings identify PPARγ as the master molecular trigger of the cardiomyocyte-to-brown/beige fat switch and demonstrate the requirement of WT1 for the conversion process.

### RhoA signaling downstream of cell–cell junctions regulates pathological myocyte-to-adipocyte conversion by controlling MRTF-A cellular localization and PPARγ activation

In order to uncover the molecular mechanisms linking defective cell–cell contacts to changes of transcriptional programs predisposing to pathological myocyte-to-adipocyte conversion, we compared the genome expression profiles of PKP2 mutant and wt CMs (Fig 5). In mutated CMs, Gene Set Enrichment Analysis (GSEA) identified significant overrepresentation of gene categories related to cell–cell and cell–matrix communication and adhesion as well as extracellular matrix remodeling (Fig 5A). Enrichment in gene sets associated with increased adipogenesis, including "adipogenesis by activated PPARγ", "classic adipogenic targets of PPARγ", and "adipogenesis early up", was likewise detected (Fig 5A and B). Since profiled cells were CMs before induction of the adipocyte switch, upregulation of the adipogenic program controlled by PPARγ was likely due to failure in PPARγ gene silencing in presence of the PKP2 mutation. PPARγ was indeed slightly upregulated (1.7-fold) in mutant CMs (Fig 5B), as shown by qRT–PCR at baseline (Fig 1F). Concordantly, analysis of histone marks of active (H3K4me3) and repressed (H3K27me3) promoters as well as active enhancers (H3K27ac) at the PPARγ locus by chromatin immunoprecipitation demonstrated significant lower levels of H3K27me3 and higher H3K4me3 and H3K27ac occupancy in mutated CMs (Fig 5C).

Remarkably, in PKP2 mutants, GSEA also revealed increased expression of genes typically upregulated in cells experiencing inhibition of the RhoA-GTPase/Rho-associated protein kinase (ROCK) pathway and undergoing cytoskeleton rearrangement (Fig 5A and D). Moreover, we found that genes known to be targets of the serum response factor (SRF) and its coregulator myocardin-related transcription factor (MRTF; Esnault et al, 2014) were likewise significantly over-represented in mutated cells (Fig 5E). These results were quite intriguing in light of recent evidence indicating that suppression of RhoA/ROCK signaling induces adipocyte differentiation and specifically brown/beige adipocyte formation via control of G-actin-regulated MRTF-A (Nobusue et al, 2014; McDonald et al, 2015). RhoA signaling mediates the dynamic control of monomeric and filamentous cytoskeleton actin (Sotiropoulos et al, 1999; Tominaga et al, 2000; Geneste et al, 2002). Monomeric G-actin can regulate the nucleus–cytoplasm shuttling of MRTF and, thereby, influence the expression of MRTF/SRF target genes (Miralles et al, 2003; Olson & Nordheim, 2010), which are required for muscle formation and identity (Balza & Misra, 2006; Mokalled et al, 2015), as well as PPARγ expression (Nobusue et al, 2014).

Since activated GTP-bound RhoA localizes at the intercellular interfaces after cell–cell contacts (Watanabe et al, 2009; Godsel et al, 2010), thus linking cell–cell adhesion to cytoskeleton remodeling and gene regulation, it appeared a strong candidate in triggering the morphological and transcriptional changes during myocyte-to-adipocyte conversion in presence of defective cell–cell contacts. We began addressing this possibility by comparing RhoA localization and the level of RhoA activity as well as cytoskeleton actin

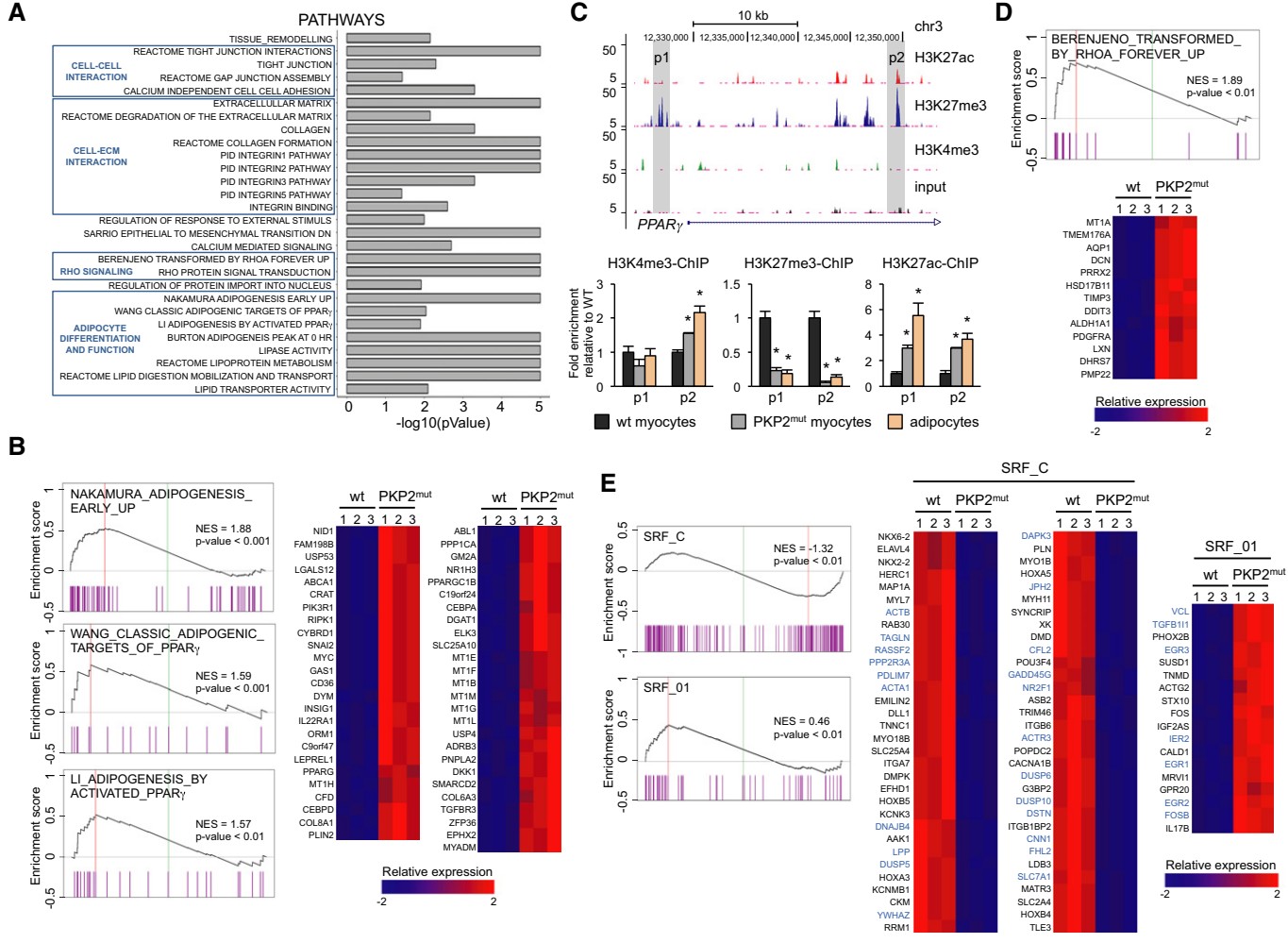

**Figure 5.  Transcriptome and histone mark analyses reveal aberrant regulation of adipogenesis and impaired RhoA- and MRTF/SRF-dependent gene programs in PKP2<sup>mut</sup> CMs.**

A    Functional annotation of genome-wide differentially expressed genes in PKP2^mut CMs with gene set enrichment analysis (GSEA).

B    GSEA of gene sets for adipogenesis (left). For each GSEA, the normalized enrichment score (NES) and *P*-value are specified. Relative expression of the "leading edge" genes of the three adipogenic GSEAs is shown in heatmaps on the right. Expression level is represented as a gradient from low (blue) to high (red).

C    Genome browser representation for H3K4me3, H3K27me3, and H3K27ac tracks at the *PPARγ* locus in wt CMs identified by ChIP-Seq (top). qPCR analyses of H3K4me3-, H3K27me3-, and H3K27ac-ChIP on p1 and p2 peaks (shaded in gray) at the *PPARγ* locus in wt and PKP2^mut CMs and human adipocytes (bottom); *n* = 2; *$P < 0.05$ vs. wt; *t*-test.

D, E   Plots of the BERENJENO_TRANSFORMED_BY_RHOA_FOREVER_UP (D) and SRF_C and SRF_01 (E) GSEAs. NES and *P*-value are shown. Relative expression of the "leading edge" genes of the GSEA is shown in the concomitant heatmaps. Direct MRTF-SRF downstream target genes are highlighted in blue. Expression level is represented as a gradient from low (blue) to high (red).

Source data are available online for this figure.

organization in wt and *PKP2* mutated CMs (Fig 6). Immunocytochemistry revealed a decreased concentration of RhoA at the plasmamembrane of mutant CMs at baseline (Fig 6A), which associated with a significant reduction in RhoA activity, assessed by ELISA as ratio "active RhoA-GTP/total RhoA" (Fig 6B). RhoA activity further declined after 24 h of culture in lipogenic milieu exclusively in *PKP2* mutants (Fig 6B), suggesting a higher responsiveness of these cells to the pro-adipogenic condition. Moreover, coincident with a reduction in RhoA activity, we observed an increase in monomeric G-actin levels in mutated CMs during the early phase of myocyte-to-adipocyte conversion, when the first

signs of disruption of cytosolic actin microfilaments (marked by Phalloidin) were visible (Fig 6C).

We next explored whether MRTF-A localization is also altered in CMs with defective cell–cell adhesion. Monomeric G-actin binds to MRTF-A and prevents it from translocating to the nucleus and regulating transcription. Initial immunohistochemistry analysis of endogenous MRTF-A in wt and *PKP2* mutated CMs before and during myocyte-to-adipocyte conversion revealed a pancellular (cytoplasmic and nuclear) localization and no clear differences between the groups (Appendix Fig S7A), suggesting a dynamic shuttling process. In order to monitor dynamic changes in MRTF-A

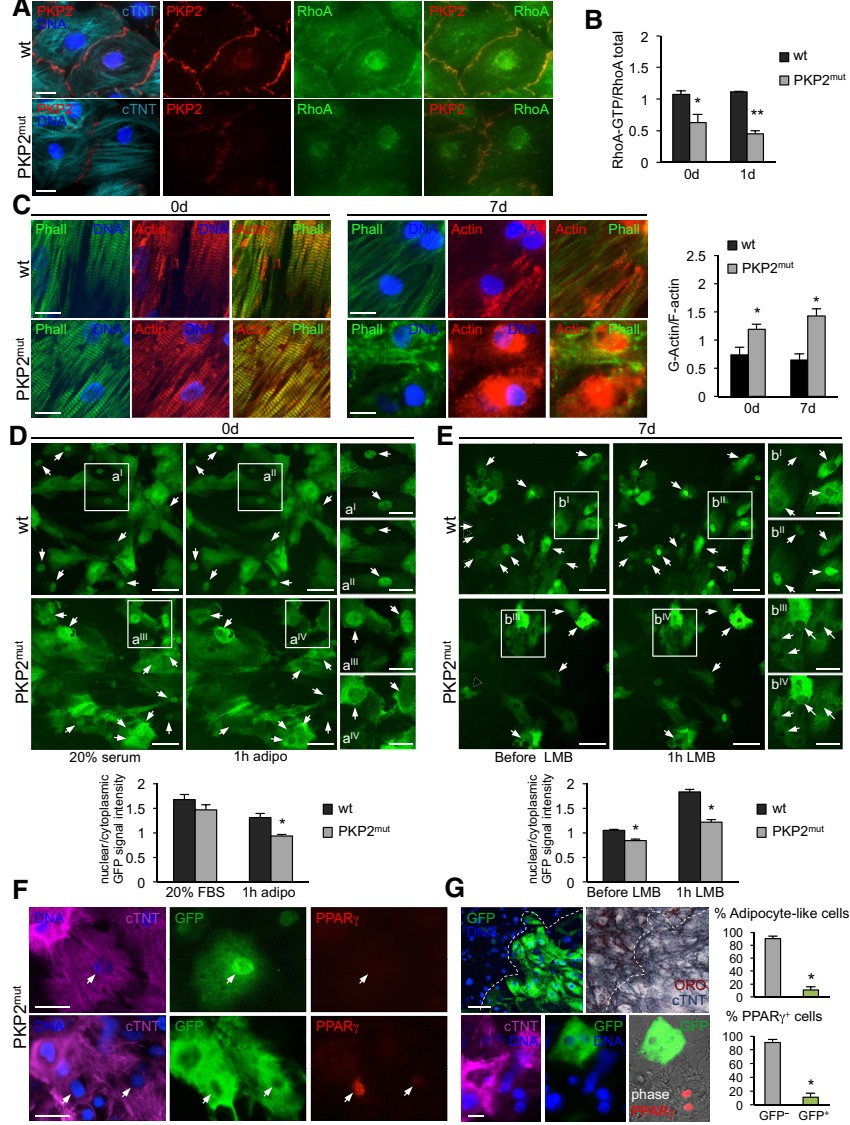

**Figure 6. Decreased RhoA signaling downstream of defective cell–cell contacts leads to reduced nuclear MRTF-A localization and derepression of PPARγ.**

A    Immunostaining of cTNT (cyan), PKP2 (red), and RhoA (green) in wt and PKP2mut CMs. Scale bars, 12.5 μm.

B    RhoA activity in wt and PKP2mut CMs at 0 and 1 day in culture; n = 3; *P < 0.05, **P < 0.001 vs. wt at the same time point; t-test.

C    Immunostaining of F-actin (Phalloidin, green) and actin (red) in wt and PKP2mut CMs at 0 and 7 days in culture. Scale bars, 12.5 μm. Bar graph shows quantification of G-actin/F-actin ratio in wt and PKP2mut CMs; N = 10 (wt) and N = 9 (PKP2mut) cells; *P < 0.01 vs. wt at the same time point; Mann–Whitney U-test.

D, E  Live imaging of MRTF-A subcellular localization in wt and PKP2mut CMs by transient expression of doxycycline-inducible MRTF-A-GFP. Images in (D) show representative CMs at d0 in 20% serum (left) and after subsequent treatment with adipogenic medium for 1 h (middle). Images in (E) show representative CMs in adipogenic medium at 7 days, before (left) and after treatment with 20 nM LMB for 1 h (middle). Small panels on the right show magnified views of the boxed areas. Arrows indicate dynamic changes of nuclear MRTF-A localization before and after corresponding treatment. Scale bars, 50 μm (left and middle panels) and 25 μm (right panels). Bar graphs show average values of nuclear/cytoplasmic ratio of GFP signal intensity in wt and PKP2mut CMs at indicated conditions; (D): N = 60 (wt) and N = 37 (PKP2mut) cells; (E): N = 105 (wt) and N = 64 (PKP2mut) cells; *P < 0.001 vs. wt at the same condition; Mann–Whitney U-test.

F    Immunostaining of cTNT (magenta), GFP (green), and PPARγ (red) in wt and PKP2mut CMs infected with MRTF-A-GFP and cultured in adipogenic medium for 7 days. Cells were treated as in (E) before fixation. Arrows indicate differential PPARγ expression dependent on levels of nuclear MRTF localization. Scale bars, 25 μm.

G    Prevention of myocyte-to-adipocyte conversion in PKP2mut CMs by continuous overexpression of MRTF-A-GFP. Immunostaining of GFP (green) and correspondent colorimetric immunodetection of cTNT (blue) in conjunction with lipid stain ORO (top) and cTNT (magenta), GFP (green), and PPARγ (red) (bottom) in PKP2mut CMs overexpressing MRTF-A-GFP cultured in adipogenic medium for 28 days. Phase-contrast image is merged with PPARγ and GFP. Scale bars, 50 μm (top), 12.5 μm (bottom). Dotted lines indicate the border separating GFP+ and GFP− (left) as well as ORO+ and ORO− (right) cells. Adjacent bar graphs indicate the percentage of adipocyte-like cells (top) or PPARγ+ cells (bottom) that were positive and negative for GFP; n = 3; N = 840 cells (top graph) and N = 306 cells (bottom graph); *P < 0.001 vs. GFP−; t-test.

Data information: All data are shown as means ± SEM.
Source data are available online for this figure.

localization in the same cell over time, we transiently expressed a doxycycline-inducible MRTF-A-GFP fused protein and treated the cells with high serum-containing medium for 30 min to induce MRTF-A nuclear translocation (Fig 6D). Shortly after switching to pro-adipogenic culture condition (1 h), *PKP2* mutants exhibited a clear reduction in nuclear GFP signal, which was conversely maintained in wt cells (Fig. 6D), suggesting a decreased nuclear shuttling of MRTF-A in mutant CMs at baseline. This was additionally confirmed at the early stage of myocyte-to-adipocyte conversion by blocking MRTF-A nuclear export with leptomycin B (LMB), which revealed a reduced nuclear MRTF-A accumulation in mutated cells at day 7 (Fig 6E). Beginning of PPARγ upregulation was detected in those cells showing the highest ratio of cytoplasmic/nuclear GFP signal (Fig 6F), suggesting that cytoplasmic sequestration of MRTF-A due to enhanced G-actin levels in *PKP2* mutant CMs could be responsible for the pathological myocyte-to-adipocyte switch by regulating PPARγ expression.

To further test whether MRTF-A localization ultimately controls PPARγ activation and thus myocyte-to-adipocyte conversion in presence of defective cell–cell contacts, we permanently applied doxycycline in the pro-adipogenic medium to induce a continuous overexpression of MRTF-A-GFP and hence a relative surplus of MRTF-A in the nucleus (Appendix Fig S7B). We then examined whether this molecular intervention could rescue or ameliorate the adipogenic switch of *PKP2* mutated CMs and influence PPARγ expression. Staining of cTNT in conjunction with ORO at 28-day culture demonstrated acquisition of the adipogenic phenotype exclusively in mutant cells lacking GFP expression, while failure to undergo adipocyte conversion was observed in the MRTF-A-GFP-overexpressing counterparts (Fig 6G). Moreover, in the latter we did not detect any PPARγ activation (Fig 6G), indicating a key role of MRTF-A dynamics and nuclear levels in determining the conversion process by modulating PPARγ expression.

### Repression of RhoA/ROCK signaling in wt CMs simulates pathological myocyte-to-adipocyte switch caused by defective cell–cell adhesion

Based on the evidence that the myocyte-to-adipocyte conversion in the presence of defective cell–cell contacts was associated with a reduction in RhoA activity, we hypothesized that pharmacological inhibition of ROCK—as the major effector of the RhoA pathway (Schofield & Bernard, 2013)—may mimic the effect of genetically impaired cell–cell adhesion. To test this, we treated wt CMs with the ROCK inhibitor Y-27632, while cells were exposed to pro-adipogenic culture conditions. Remarkably, after only 14-day treatment, disruption of cell–cell junctions was evident in the majority of CMs, as illustrated by weaker and interrupted plakophillin expression at the plasmamembrane (Fig 7A), suggesting a positive feedback regulation of RhoA/ROCK signaling on cell–cell contact stability. Furthermore, exposure to Y-27632 for 28 days induced a severe remodeling of the actin cytoskeleton with nuclear shuttling of WT1 and activation of PPARγ expression (Fig 7B), which ultimately correlated with the appearance of few lipid-filled adipocyte-like cells, as shown by ORO staining (Fig 7C). Although the efficacy of this pharmacological approach appeared low, the overall ability of some CMs to convert into adipocytes upon ROCK inhibition indicates a critical function of RhoA/ROCK signaling in integrating cell–cell junctional

structures with cytoskeletal organization to influence localization and activation of proteins that control transcriptional programs affecting CM state. Moreover, it reinforces the hypothesis that subsets of CMs exist, which are permissive to undergo adipocyte conversion more readily than others, possibly due to a shared common developmental origin with cardiac adipocytes.

### Genetic perturbation of RhoA-mediated cytoskeleton remodeling induces myocyte-to-adipocyte switch

To further verify the importance of RhoA/ROCK-dependent cytoskeleton remodeling in driving pathological myocyte-to-adipocyte switch, we analyzed whole-exome sequencing data from ARVC patients lacking mutations in classical disease-causing genes. We reasoned that if this mechano-signaling is relevant in ARVC, then genes belonging to the RhoA/cytoskeletal pathway may be targets of pathogenic mutations. Strikingly, in one of four cases we identified a heterozygous nonsense mutation (c.1729C>T; p.R577*) in *MYH10*. This gene encodes the non-muscle myosin IIB (NMIIB) of the actomyosin cytoskeleton, which has been reported to regulate both actin dynamics (Rex *et al*, 2010) and accumulation of active RhoA-GTP at the adherens junctions (Priya *et al*, 2015). Remarkably, among all genes that have been associated with ARVC, *MYH10* showed the maximum value of 1 for the ExAC index pLI (which indicates the probability that a gene is intolerant to a loss-of-function mutation—see Materials and Methods and Appendix Table S1), suggesting a high likelihood for the novel R577*-*MYH10* mutation to be disease causing. We generated iPSCs harboring the R577*-*MYH10* mutation (Appendix Fig S8A–G) and examined the phenotype of the derived CMs before and after exposure to lipogenic milieu (Fig 7D–J). Western blot analyses revealed a ~50% reduction in NMIIB protein, while transcript levels were unchanged (Appendix Fig S8H and I). Already at baseline, a reduced accumulation of NMIIB at the cell–cell junctions was evident in *MYH10* mutated CMs and correlated with a decreased plasmamembrane concentration of RhoA (Fig 7D and E). Interestingly, compromised cell–cell adhesion was also discernable (Fig 7E and Appendix Fig S8J). These changes associated with elevated levels of cytosolic G-actin and impaired nuclear shuttling of MRTF-A (Fig 7F and G). Moreover, we measured a basal increase in *PPARγ* and *PRDM16* transcripts, which further progressively augmented during culture in lipogenic milieu (Fig 7H). Similarly to CMs with defective PKP2 levels, a subset of *MYH10* mutants exhibited nuclear translocation of WT1 and activation of PPARγ when cultured in pro-adipogenic conditions (Fig 7I) and ultimately converted into adipocytes (Fig 7J).

Together, these findings indicate an unprecedented function of NMIIB as canonical RhoA effector in regulating CM identity and support the pivotal role of RhoA signaling in triggering cardiomyocyte-to-brown/beige adipocyte conversion in response to adipogenesis-inducing cues.

## Discussion

While it is widely accepted that mechanical cues contribute to cell fate decisions during differentiation of stem cells (McBeath *et al*, 2004), little is known about how mechanical inputs at cell–cell or

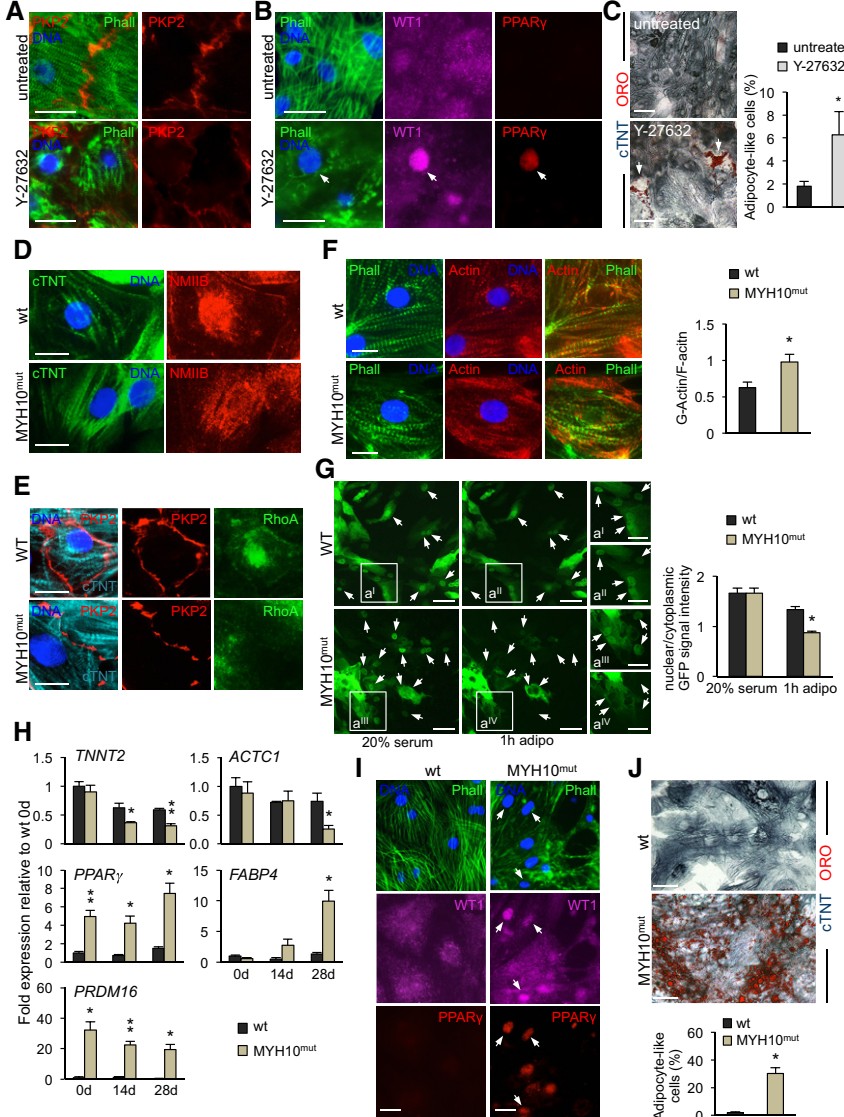

**Figure 7.  Pharmacological and genetic perturbation of RhoA-mediated cytoskeleton remodeling induces myocyte–adipocyte conversion.**

A, B    Immunostaining of F-actin (Phalloidin, green) and PKP2 (red) in wt CMs treated and non-treated with 30 μM ROCK inhibitor Y-27632 for 14 days (A). Immunofluorescence analysis of WT1 (magenta), F-actin (Phalloidin, green), and PPARγ (red) in wt CMs treated and non-treated with 30 μM Y-27632 for 28 days. Arrow indicates a cell with severe remodeling of the actin cytoskeleton, nuclear shuttling of WT1, and PPARγ expression (B). Scale bars, 25 μm.

C    Immunodetection of cTNT (blue) and lipid stain ORO (red) in wt CMs untreated and treated with 30 μM Y-27632 for 28 days. Scale bars, 50 μm. Bar graph shows the percentage of adipocyte-like cells in treated and untreated conditions; $n = 3$; $N = 600$ (untreated) and $N = 592$ (Y-27632) cells; *$P < 0.05$ vs. untreated cells; $t$-test.

D, E    Immunostaining of cTNT (green) and NMIIB (red) (D) or cTNT (cyan), PKP2 (red), and RhoA (green) (E) in wt and MYH10$^{mut}$ CMs. Scale bars, 12.5 μm.

F    Immunostaining of F-actin (Phalloidin, green) and actin (red) and quantification of G-actin/F-actin ratio in wt and MYH10$^{mut}$ CMs. Scale bars, 12.5 μm. $N = 11$ (wt) and $N = 13$ (MYH10$^{mut}$) cells; *$P < 0.05$ vs. wt; Mann–Whitney $U$-test.

G    Live imaging of MRTF-A subcellular localization in wt and MYH10$^{mut}$ CMs by transient expression of doxycycline-inducible MRTF-A-GFP. Shown are representative myocytes at d0 in 20% serum (left) and after subsequent treatment with adipogenic medium for 1 h (middle). Small right panels show magnified views of the boxed areas. Arrows indicate dynamical changes of nuclear MRTF-A localization. Scale bars, 50 μm (left and middle) and 25 μm (right). Bar graph presents quantification of nuclear/cytoplasmic ratio of GFP signal intensity in wt and MYH10$^{mut}$ CMs; $N = 55$ (wt) and $N = 53$ (MYH10$^{mut}$) cells; *$P < 0.001$ vs. wt; Mann–Whitney $U$-test.

H    qRT–PCR analysis of myocytic and adipocytic genes in wt and MYH10$^{mut}$ CMs over time in culture; $n = 3$; *$P < 0.05$, **$P < 0.01$ vs. wt at the same time point; $t$-test.

I    Immunostaining of F-actin (Phalloidin, green), WT1 (magenta), and PPARγ (red) in wt and MYH10$^{mut}$ CMs at 28 days of culture. Arrows indicate cells with severe remodeling of the actin cytoskeleton, nuclear WT1, and PPARγ expression. Scale bars, 25 μm.

J    Colorimetric immunodetection of cTNT (blue) and lipid stain ORO (red) in wt and MYH10$^{mut}$ CMs at 28 days of culture. Scale bars, 50 μm. Bar graph shows the percentage of adipocyte-like cells in wt and MYH10$^{mut}$ conditions; $n = 3$; $N = 567$ (wt) and $N = 520$ (MYH10$^{mut}$) cells; *$P < 0.05$ vs. wt ($t$-test).

Data information: All data are shown as means ± SEM.
Source data are available online for this figure.

cell–matrix adhesions translate into intracellular signals to maintain cell identity or direct fate switch. Here we describe an adhesion-linked signaling and transcriptional regulatory pathway that mediates myocyte-to-adipocyte lineage conversion (Fig 8), thereby providing a mechanism by which mechanical signals through cell–cell contacts integrate with soluble cues and translate into a change in cell fate. Using human iPSCs with defective cell–cell adhesion due to a *PKP2* frameshift mutation or impaired RhoA signaling caused by a nonsense *MYH10* mutation, we show that cell–cell contact-mediated mechanosensing plays a pivotal role in controlling CM identity through a series of interwoven, reinforcing pathways that converge on the engagement of a RhoA/MRTF/SRF-signaling circuit. When impaired, CMs become primed to switch to brown/beige adipocytes in response to adipogenesis-inducing signals acting on this circuit. Moreover, by tracking the fate of murine cardiovascular progenitors expressing Isl1 and Wt1, we provide evidence for an intimate developmental relationship between CMs and AV groove adipocytes. Accordingly, we demonstrate that a single Isl1[+]/Wt1[+] progenitor can give rise to both CMs and fat cells *in vitro*, which ultimately suggests related mechanisms of determination between the two lineages.

Our data show that in human developing CMs cell–cell contacts at the intercalated disk connect to remodeling of the actin cytoskeleton by regulating RhoA-ROCK signaling to maintain an active MRTF/SRF transcriptional program. By controlling expression of muscle-specific and adipogenesis-related genes as well as cytoskeleton, intercellular junction, and extracellular matrix genes, MRTF/SRF sustains cardiac muscle identity and function, while inhibiting adipogenic commitment (Fig 8). Reduction in RhoA-ROCK signaling initiated by defects in cell–cell adhesion poises CMs to convert to

adipocytes in response to external adipogenesis-inducing cues, which act as feed-forward mechanism by further downregulating the RhoA-MRTF/SRF pathway. Indeed, standard pro-adipogenic cocktails, such as the one used in this study, contain IBMX, dexamethasone, and insulin. Both IBMX and dexamethasone inhibit RhoA/ROCK activity by activating protein kinase A (Petersen *et al*, 2008). The fact that a lipogenic milieu provoked cardiomyocyte-to-adipocyte conversion only in cells with impaired cell–cell junctions or RhoA signaling and that additional ROCK inhibition was necessary to trigger the cell fate change in wt CMs indicates that, in absence of a permissive environment, such as altered mechanosensing, this feed-forward mechanism does not become activated. Moreover, 4-week treatment with the ROCK inhibitor Y-27632 resulted in cardiomyocyte-to-adipocyte switch in a limited number of wt cells, suggesting that only some CMs are primed for this cell fate switch, presumably because of permissive transcriptional or epigenetic states acquired during development. In line with this, it has recently been shown that ROCK inhibition in the developing heart, but not after birth, leads to fibrofatty replacement of CMs in the RV of adult mice, which is associated with markedly disorganized myocardial structures and cell–cell junctions, as well as increased PPARγ levels, pointing toward adipogenic phenotypic changes (Ellawindy *et al*, 2015). Moreover, our results, indicating that a subset of cardiac muscle and epicardial fat cells share a common developmental origin and that CMs differentiated from mESC-derived Isl1[+]/Wt1[+] clones are highly responsive to adipogenic stimuli, suggest that CMs emerging from Isl1[+]/Wt1[+] progenitors could be those more prone to a myocyte-to-adipocyte switch when exposed to adipogenesis-inducing cues, though this remains to be examined *in vivo*.

While MRTF/SRF signaling has been shown to be essential for CM differentiation and function (Niu *et al*, 2005; Parlakian *et al*, 2005; Mokalled *et al*, 2015) and to inhibit differentiation of preadipocytes and multipotent mesenchymal progenitor cells into adipocytes (Mikkelsen *et al*, 2010; Nobusue *et al*, 2014; McDonald *et al*, 2015), no evidence existed for a role in controlling cell fate switch between these two mesodermal lineages. Our work demonstrates that overexpression of MRTF-A in human *PKP2* mutated CMs prevents their conversion into adipocytes in response to adipogenesis-inducing signals by hindering PPARγ activation. Thus, it will be important to investigate whether it can be used therapeutically in ARVC. Interestingly, it has been reported that RhoA-GTPase is an essential modulator of insulin-like growth factor 1 (IGF1) signals that direct the adipogenesis–myogenesis cell fate decision of common mesenchymal precursors *in vivo* and *in vitro* (Sordella *et al*, 2003). Whether RhoA activity controls adipogenesis–myogenesis decision through involvement of MRTF/SRF has not been examined, however.

Recent evidence suggests that IGF1R signaling also governs epicardial adipose tissue formation in the context of myocardial injury in mice by redirecting the fate of Wt1[+] lineage cells (Zangi *et al*, 2017). Wt1 expression seems to be required for such process (Zangi *et al*, 2017). It is relevant to mention that, differently from humans, the mouse heart has very little fat during homeostasis and only upon injury does some degree of relaxation in the negative control of adipogenesis occur (Liu *et al*, 2014; Yamaguchi *et al*, 2015). Interestingly, Wt1 expression is restricted only to cells of the epicardium, the outer mesothelial layer of the heart, and completely absent in CMs (Zhou *et al*, 2008). Our immunocytochemical and

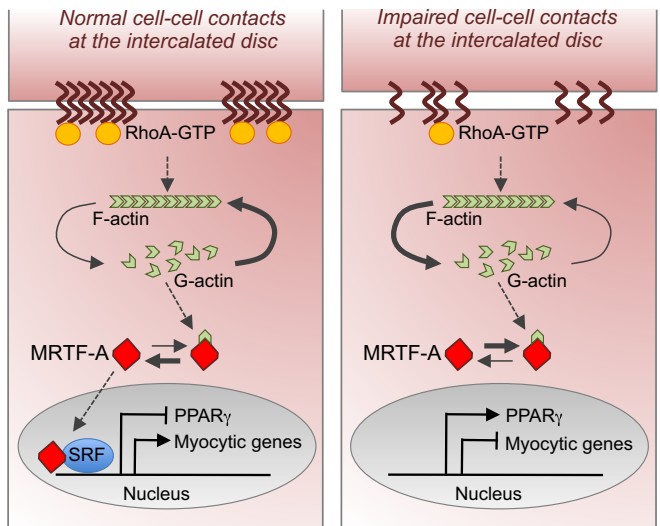

**Figure 8. Model describing cell–cell adhesion-dependent regulation of cardiomyocyte-to-adipocyte lineage conversion.**

Impaired intercellular contacts at the intercalated disk of developing cardiac myocytes lead to a reduction in RhoA-GTPase activity and remodeling of actin cytoskeleton resulting in increased level of cytosolic G-actin. G-actin binding to MRTF prevents its nuclear translocation, resulting in repression of the MRTF/SRF-regulated myogenic lineage commitment and activation of the alternative adipocytic gene program triggered by PPARγ.

immunohistochemical analyses of human iPSC-derived CMs and adult human heart tissue indicate that high levels of WT1 are normally detectable in the cytoplasm of human cardiac muscle, confirming previous studies in developing human hearts (Ambu *et al*, 2015). Remarkably, we observed a nuclear translocation of WT1 protein occurring during the early phase of cardiomyocyte-to-adipocyte conversion in *PKP2 and MYH10* mutant cells as well as wt CMs treated with the ROCK inhibitor Y-27632, suggesting an involvement of cell contact-mediated RhoA signaling in resolving activation of WT1 as a transcription factor in these cells. Further studies will be needed to address the precise mechanism by which RhoA/ROCK pathway controls WT1 subcellular localization. Furthermore, our overexpression and loss-of-function studies in wt human iPSC-CMs and in adult CMs of $Myh6^{Cre/+}$;$R26^{mTmG/+}$ mice indicate that WT1, although not sufficient to provoke the myocyte-to-adipocyte switch, is essential for the conversion process induced by PPARγ. Additionally, we measured a physical co-interaction of WT1 and PPARγ in the nucleus of myocyte–adipocyte converting cells, suggestive of a cooperative role of the two transcription factors in regulating the adipogenic program.

Our expression profiling results and histone modification analysis of the *PPARγ* locus indicate an aberrant ectopic activation of PPARγ and the adipocytic program in *PKP2* mutant CMs before any exposure to adipogenesis-inducing signals. This finding, together with the fact that only a fraction of mutated CMs could convert to adipocytes in our experimental conditions, supports the concept that faulty mechanical sensing due to defects in cell–cell contact may prevent adipogenic gene silencing and faithful cardiomyocytic commitment of a bipotent myo-adipo progenitor cell during cardiogenesis. As result, the differentiated CM progeny will be permissive to switch to the alternative adipocytic fate when exposed to appropriate stimuli.

In agreement with our work, previous studies have reported exaggerated lipogenesis and abnormal PPARγ activation in iPSC-derived CMs from ARVC patients carrying *PKP2* mutations when exposed to adipogenesis-inducing cues (Caspi *et al*, 2013; Kim *et al*, 2013; Ma *et al*, 2013). However, a stable lineage switch of CMs to adipocytes was not resolved in these studies. By applying culture settings that mimic the mechanical strain of the adult heart, we provide the first evidence of a direct lineage conversion that naturally occurs under pathological conditions in human cells.

Taken together, we have identified a cell–cell contact-driven transcriptional rheostat that controls human CM identity and when impaired predisposes CMs to convert into brown/beige adipocytes in response to adipogenesis-inducing soluble cues. This study sheds important light on how cell–cell junctional architecture, cytoskeleton organization, and transcriptional regulation are coupled to regulate cellular identity. Thus, it provides novel insights on how mechanical signals in CMs can control cell fate switch by feeding into the regulation of core pathways.

# Materials and Methods

### Generation, culture, and *in vitro* differentiation of human iPSCs

Dermal keratinocytes from PKP2^mut and MYH10^mut ARVC patients and healthy controls were isolated as previously described (Aasen

*et al*, 2008) and reprogrammed into iPSCs using Sendai viruses coding for *OCT4, SOX2, KLF4,* and *c-MYC*. All recruitment and consenting procedures were done under institutional review board-approved protocols of both the Klinikum rechts der Isar, Technical University of Munich, and the Addenbrooke's and Papworth Hospitals, Cambridge University Health Partners. Written informed consent was obtained from affected patients and healthy volunteers.

Spontaneous differentiation of iPSCs into cells of all three germ layers was induced by embryoid body (EB) formation, as previously reported (Moretti *et al*, 2010). Neural differentiation was achieved by growing EBs in suspension for the first 7 days and then as adherent EBs on laminin-coated dishes for further 2 weeks.

iPSC-derived CMs were obtained using PSC Cardiomyocyte Differentiation Kit (Life Technologies). Two- to three-month-old CM cultures were dissociated into single cells using 480 U/ml collagenase type II (Worthington) and purified with PSC-derived Cardiomyocyte Isolation Kit, human (Miltenyi). CM purity assessed by flow cytometry reached 97%. Single CMs were then plated on fibronectin-coated 50-kPa dishes (Matrigen) as monolayer, subjected to 1 Hz electrical pacing, thus imitating the laminar sheet-like architecture, the stiffness, and the beating-induced cellular tension of the adult heart tissue. Treatment with adipogenic medium (Lonza) containing insulin, dexamethasone, indomethacin, and IBMX was initiated 3 days later. Samples were collected at d0, d14, d28, and 8 weeks in culture in adipogenic medium for immunohistochemical and molecular biological analysis. All experiments were performed using two clones for each iPSC line. Wild-type cells were obtained from three independent control iPSC lines from three unrelated healthy individuals.

Human iPSC-derived brown/beige adipocytes were differentiated as previously described (Guenantin *et al*, 2017).

### Teratoma formation

Six-week-old severe combined immunodeficient (SCID) mice (Charles River Laboratories) were anesthetized, and ~1 million iPSCs were inoculated beneath the kidney capsule. Teratomas were harvested 6 weeks post-injection, paraffin embedded, and stained with hematoxylin and eosin for the histological determinations.

### Exome sequencing, genomic sequencing, and karyotype analysis

Genomic DNA was isolated using Genomic DNA Purification Kit (Gentra Systems). Direct sequencing of the five potentially causative desmosomal genes (*PKP2*, *DSP*, *JUP*, *DSG2,* and *DSC2*) was performed at the UCL Centre for Inherited Cardiovascular Disease with coverage of all exons as well as flanking intronic regions and revealed the previously described heterozygous frameshift mutation (c.1760delT; p.V587Afs*655) in PKP2 (Te Riele *et al*, 2013) in one patient. For exome sequencing, SureSelect Human All Exon 50 Mb kit (Agilent) was applied for in-solution enrichment of exonic regions followed by sequencing as 100 base-pair (bp) paired-end runs on a HiSeq2500 (Illumina). Reads were aligned to the human genome assembly hg19 (UCSC Genome Browser) with Burrows-Wheeler Aligner (BWA, v.0.5.87.5), and detection of genetic variation was performed using SAMtools (v0.1.18), PINDEL (v0.2.4t), and ExomeDepth (v1.0.0). Variant annotation was performed applying custom Perl scripts, including information about known transcripts (UCSC

genes and RefSeq genes), known variants (dbSNP v135), type of mutation, and amino acid change (where applicable). The obtained variants were then inserted into the Helmholtz Zentrum München database, including 10,900 exomes. All synonymous and intronic (other than canonical splice sites) variants were excluded, and quality control filters were applied as well (variant quality ≥30 and mapping quality ≥50). Genetic variants were interrogated in the 1000 Genomes project (www.1000genomes.org) and the Exome Aggregation Consortium (ExAC) browsers (http://exac.broadinstitute.org) and Sanger validated. Predicted functional effect of a coding variant was surveyed using Polyphen2 (http://genetics.bwh.harvard.edu/pph/), SIFT (http://sift.jcvi.org/), and Combined Annotation Dependent Depletion (CADD; http://cadd.gs.washington.edu/). Exome sequencing yielded 13 Gb of mappable sequences, and 98% of the targeted sequences were covered at least 20-fold with a mean coverage of at least 139×. Bioinformatics filtering of variants was based on recessive and dominant inheritance patterns. Prioritization of recessive protein-altering variants with a minor allele frequency 0.001% in the Exome Aggregation Consortium (ExAC) Browser (v.0.3.1) or an internal database containing 10,900 control exomes identified 18 bi-allelic variants that were considered unlikely to be causative for the observed ARVC phenotype. Considering a dominant effect, the index subject's WES data were quested for protein-altering sequence variations with a minor allele frequency 0.0001% in the ExAC Browser or in the internal database. Among 211 heterozygous variants, a phenotype-based search identified nonsense variant in MYH10 (c.1729C>T, p.R577*). The variant was not present among 10,900 internal exomes and 60,706 exomes in the ExAC database. The probability of loss-of-function intolerance (pLI) in ExAC browser reaches the maximum value. The amino acid change involves a highly conserved amino acid residue (CADD score 38).

Genotype of the patient iPSCs was confirmed by sequencing (Eurofins MWG Operon) on genomic DNA from iPSCs with primers corresponding to the mutated region located in exon 8 of the *PKP2* gene or exon 15 of the *MYH10* gene. Primer sequences are provided in Appendix Table S2.

Karyotyping of the iPSC lines was performed at the Institute of Human Genetics of the Technical University of Munich using standard methodology.

## Generation, culture, and differentiation of mouse ESCs

$Isl1^{Cre/+};R26^{YFP/+}$ mouse line was used to isolate ESCs as previously described (Nichols *et al*, 2009). ESCs were maintained in mESC medium and differentiated in EBs as previously reported (Moretti *et al*, 2006). Single cells were dissociated from 4-day-old EBs and sorted for YFP expression using flow cytometry. $YFP^+$ cardiac progenitors were either differentiated into adipocytic and cardiomyocytic lineage or plated on mitomycin C-treated cardiac mesenchymal feeder cells (CMC) for clonal expansion. For adipogenic differentiation, cells were plated in mESC medium supplemented with 5 μM insulin (Sigma-Aldrich) and 0.1 μM dexamethasone (Sigma-Aldrich) [mESC + 2F] on 0.1% gelatin-coated plates and cultured for 4 weeks. For myogenic differentiation, cells were reaggregated in mESC medium and collected for analysis 6 days later. Progenitors seeded for clonal experiments were cultured for 6 days on CMCs, and the resulting clones were dissociated into single cells and then divided into three parts.

One part was differentiated in adipogenic conditions as decribed above. The second part was differentiated in myogenic conditions (Moretti *et al*, 2006). The third part was used for RT–PCR analysis.

## Reverse transcription PCR (RT–PCR) and quantitative real-time PCR (qRT–PCR)

Total mRNA was isolated from keratinocytes, iPSCs, CMs, iPSC-derived brown/beige adipocytes, and cardiac progenitors using the Stratagene Absolutely RNA kit, and 1 μg was used to synthesize cDNA with the High-Capacity cDNA Reverse Transcription kit (Applied Biosystems). Gene expression was quantified by qRT–PCR using 1 μl of the RT reaction and the Power SYBR Green PCR Master Mix or Taqman assays (Applied Biosystems). Gene expression levels were normalized to *GAPDH*. A list of the primers and Taqman probes is provided in Appendix Table S2.

## Genome-wide transcriptional profiling and Gene Set Enrichment Analysis—GSEA

Microarray expression profiling was performed on pooled mRNA isolated from iPSC-derived CMs obtained from three independent differentiation experiments. The GeneChip microarray processing was performed by IMGM Laboratories (Munich) according to the manufacturer's protocols. The amplification and labeling were processed with 100 ng starting RNA. For each sample, 600 ng single-stranded DNA was labeled and hybridized to the SurePrint G3 Human GE Microarray chips. Fluorescent signal intensities were detected with Scan Control A.8.4.1 software (Agilent Technologies) on the Agilent DNA Microarray Scanner and extracted by the Feature Extraction 10.7.3.1 software (Agilent Technologies). The software tools Feature Extraction 10.7.3.1, GeneSpring GX 11.5.1 (Agilent Technologies), and Spotfire Decision Site 9.1.2 (TIBCO) were used for quality control and statistical analysis. Agilent Sure-Print G3 Human GE Microarrays (8 × 60 K) raw intensity data were imported in R statistical environment using limma package (Ritchie *et al*, 2015) for background subtraction, quantile normalization, and log2 transformation of the signal values. Linear modeling approach, empirical Bayes statistics with Benjamini and Hochberg multiple testing correction was utilized to obtain genes whose fold change between comparisons was > 1.5 with a *P* value cutoff of < 0.05. Gene Set Enrichment Analysis was performed using GSEA (Subramanian *et al*, 2005; 1.000 permutations; genes ranked according to PKP2$^{mut}$ vs. wt −log *P*-value * sign FC).

## Chromatin Immunoprecipitation Sequencing (ChIP-Seq) and data processing

$3 \times 10^4$ 2-month-old WT, PKP2$^{mut}$ CMs, and *in vitro* differentiated human adipocytes (Lonza) were cross-linked for 15 min in 1% formaldehyde (Thermo Scientific). The reaction was stopped with 0.125 M glycine. Samples were resuspended in RIPA buffer and divided into three parts (10,000 cells each). Chromatin was sheared to 150–500 bp fragments with a Biorupter Plus sonication device (Diagenode) and precipitated using 1 μg anti-H3K4me3 (Diagenode), anti-H3K27me3 (Abcam), or anti-H3K27ac (Millipore) overnight at 4°C. Next day, chromatin–antibodies complexes were

incubated with Dynabeads Protein A (Thermo Scientific) for 2 h at 4°C. Elution of the precipitated chromatin was done at 68°C for 2 h and reverse cross-linking at 65°C for 5 h. ChIP-DNA was purified by treatments with Proteinase K (Thermo Scientific), RNase A (Thermo Scientific), and standard phenol:chloroform: isoamylalcohol extraction and amplified using the SeqPlex amplification Kit (Sigma-Aldrich). ChIP assay buffers are described (Weishaupt & Attema, 2010).

Quality and quantity of the ChIP-DNA were assessed using TapeStation (Agilent Technologies). Library preparation was performed with the TruSeq ChIP sample preparation kit (Illumina Inc.). End-repair, adapter ligation, and 10 cycles of PCR were done on 20 ng starting material. Pippin Prep (SageScience) was used for size selection of the ChIP libraries with a range of 270–350 bp. Final library was quality checked using the Caliper LabChip GX DNA 1K Assay and quantified using a picoGreen assay (Invitrogen). ChIP library was sequenced as 100 pb paired-end run on a HiSeq2500 (Illumina Inc.).

The quality assurance, pre-processing, and peak calling workflows were implemented in Galaxy (Goecks et al, 2010). Quality assurance was performed using the FastaQC (http://www.bioinformatics.babraham.ac.uk/projects/fastqc). Reads were aligned against the Homo sapiens genome using bowtie (Langmead et al, 2009). BAM files were filtered removing unmapped reads, multi-mapping reads, and duplicate reads. Finally, peaks were called with MACS v2.1.0. (Zhang et al, 2008). RPGC normalized coverage (bigwig) and bedGraph files are created at several steps of the workflow to visualization of the tracks in the UCSC Genome Browser http://genome.ucsc.edu.

### Immunoprecipitation and Western blotting

For co-IPs, nuclear extracts were prepared from wt CMs transduced with N-terminally HA-tagged-PPARγ and NLS-WT1 14 days after infection and culturing in adipogenic medium using NE-PER Nuclear Extraction Kit (Thermo Fisher) according to manufacture's protocol. 200 μg nuclear lysate was incubated with 2 μg of anti-Wt1 or anti-HA-tag antibody overnight at 4°C, followed by 1-h incubation with Protein G Dynabeads (Thermo Fisher). Immunoprecipitates were extensively washed with PBST, dissolved in Laemmli buffer, and analyzed by Western blotting. Rabbit IgG was used as a negative control. Western blot analysis of samples taken before (40 μg of nuclear extract, input) and after immunoprecipitation was performed with an anti-Wt1 and anti-PPARγ antibodies, followed by light chain and heavy chain specific secondary HRP-conjugated antibodies, respectively. Western blotting was performed using standard protocols. Antibodies used for IPs and Western blotting are provided in Appendix Table S3. Protein band intensities were quantified with ImageJ.

### Immunohistological analysis

Adult mouse hearts and embryos were fixed in 4% paraformaldehyde (PFA) for 2 h at 4°C, frozen in OCT, and sectioned at 8 μm. Human myocardium biopsies from ARVC and non-ARVC patients were fixed in 4% PFA for 2 h at 4°C, dehydrated, and paraffin embedded. 8-μm sections were rehydrated, and antigen-retrieval was performed in the citrate buffer (VECTOR Laboratories). Cells

and EBs were fixed with 4% PFA for 15 min at RT. Samples were blocked with 10% FBS for 90 min and subjected to a specific immunostaining at 4°C overnight using primary antibodies as listed in Appendix Table S3. Phalloidin (Thermo Fisher Scientific) was used to stain F-actin. AlexaFluor488-, AlexaFluor-594-, AlexaFluor-647-conjugated secondary antibodies (Thermo Fisher Scientific or Jackson Immunoresearch) specific to the appropriate species were used. Nuclei were detected with 1 μg/ml Hoechst 33258. For immunoperoxidase staining, the VECTASTAIN ABC system (VECTOR Laboratories) was used. Lipids were stained with BODIPY (Thermo Fisher Scientific) or ORO.

Microscopy was performed with a DMI6000-AF6000 or a SP8 II confocal laser-scanning Leica microscope. Images were assigned with pseudo-colors and processed with ImageJ or Photoshop.

For G-actin quantification, different images were calculated in which the phalloidin signal was subtracted from the α-actin signal. The fluorescence intensity over representative cells was quantified in this difference image and in the corresponding phalloidin image, and a ratio representing monomeric/filamentous actin was calculated for each cell.

### Glycerol release

To measure lipolysis activity, 8-week-old wt and PKP2$^{mut}$ CMs were starved overnight and then incubated in DMEM supplemented with fatty acid-free BSA alone or with 10 μM forskolin (Millipore) for 4 h. The culture medium was collected to measure glycerol using the free glycerol reagent (Sigma), and cells were then lysed in RIPA buffer (Sigma) for protein isolation. Glycerol (μg) was normalized to protein content (mg).

### Measurement of active RhoA

Two-month-old wt and PKP2$^{mut}$ CMs were plated on fibronectin-coated plates. Four days after plating (d0), adipogenic medium was applied for 1 day (d1). Cells at d0 and d1 were lysed and subjected to measurement of total and GTP-bound RhoA levels using a total RhoA ELISA and RhoA activation G-LISA kit, respectively (Cytoskeleton). Active RhoA was calculated as the RhoA-GTP/total RhoA ratio.

### NLS-WT1, PPARγ, and MRTF-A-GFP overexpression and shRNA silencing of WT1 in CMs

Human PPARγ and NLS-WT1 cDNA were cloned into the lentiviral plasmid pRRLsin18.PPT.PGK.IRES.GFP. In NLS-WT1, the coding sequence of WT1 is preceded by SV40 large T antigen NLS. PPARγ coding sequence is fused N-terminally with HA-tag. MRTF-A cDNA C-terminally fused to GFP was inserted into the doxycycline-inducible lentiviral plasmid pInducer20 (Meerbrey et al, 2011); the neomycin resistance was replaced by puromycin cassette. For knockdown of WT1, shRNA oligo was cloned into the doxycycline-inducible lentiviral plasmid pGLTR-GFP (provided by Lukas Huber, Biocenter, Innsbruck Medical University, Innsbruck, Austria). As a control, an shRNA non-targeting oligo (scrambled) was used. Lentiviruses were produced as previously described (Dorn et al, 2015). Wt CMs were transduced with GFP control, NLS-WT1, PPARγ and GFP, or PPARγ and NLS-WT1 lentiviruses, and adipogenic treatment

was initiated 3 days after infection. Samples for RNA and immuno-stainings were collected before infection (d0) and after 7, 12, 14, 21, and 28 days of culturing in adipogenic medium. Expression of MRTF-A and shWT1 or scramble was induced with 1 and 0.1 μg/ml doxycycline (AppliChem), respectively. PKP2$^{mut}$ CMs were infected with MRTF-A-GFP and treated with adipogenic medium and doxycycline for 28 days.

### Live cell imaging of MRTF-A-GFP localization

To visualize MRTF-A subcellular localization at d0 wt, PKP2$^{mut}$ and MYH10$^{mut}$ CMs were transduced with MRTF-A-GFP and its expression was induced for 2 days with doxycycline 3 days after infection. Two days later, the cells were starved in 0.5% FBS for 12 h and imaged after stimulation with 20% FBS and subsequent incubation with adipogenic medium for 1 h. To visualize MRTF-A subcellular localization at day 7, wt and PKP2$^{mut}$ CMs transduced with MRTF-A-GFP were treated with adipogenic medium 3 days after infection and MRTF-A-GFP expression was induced for 2 days with doxycycline at d3 of adipogenic culture. At day 7, cells were imaged before and after application of 20 nM LMB (Calbiochem) for 1 h. For immunofluorescence analysis, cells at 7 days were treated with 20 nM LMB for 1 h before fixation. To quantify the MRTF-A subcellular localization, nuclear/cytoplasmic ratio of the GFP average signal intensity of the cells imaged before and after treatment was calculated.

### Transgenic mice and AVV9-mediated transduction

*Isl1$^{Cre/+}$*, *Isl1$^{MerCreMer/+}$* (provided by Sylvia Evans, University of California-San Diego, La Jolla, CA), *Wt1$^{CreERT2/+}$*, *Myh6$^{Cre/+}$*, *R26$^{YFP/+}$, and R26$^{mTmG/+}$* (Jackson Laboratory) mice were used for lineage-tracing experiments. To induce Cre activity of *Isl1$^{MerCreMer/+}$* and *Wt1$^{CreERT2/+}$*, tamoxifen (Sigma-Aldrich) was injected intraperitoneally into pregnant females at E7.5 at 3 mg per 40 g body weight.

Recombinant AAV9-NLS-WT1 and AAV9-PPARγ vectors were produced in U293 cells, purified with standard cesium sedimentation, and tittered as previously described (Ziegler *et al*, 2013). For cardiomyocyte-specific transduction, *Myh6$^{Cre/+}$;R26$^{mTmG/+}$* mice received $2.5 \times 10^{12}$ virus particles in the tail vein.

### Statistical analysis

Data that passed tests for normality were analyzed with the use of *t*-tests for equal (Student's *t*-test) or unequal (Welch's *t*-test) variances. For data that were not normally distributed, a Mann–Whitney *U*-test was applied. *P*-values of less than 0.05 were considered statistically significant. All data are shown as means ± SEM.

### Source data

Microarray expression data have been deposited in the Gene Expression Omnibus (GEO) under the accession code GSE98525 (https://www.ncbi.nlm.nih.gov/geo/query/acc.cgi?&acc=GSE98525). Source data for Figures, Expanded View Figures, and Appendix Figures have been provided.

**Expanded View** for this article is available online.

## Acknowledgements

We thank the ARVC patients and the healthy volunteers who provided us with skin biopsies for the reprogramming. We would like to acknowledge Birgit Campbell and Christina Scherb for their technical assistance in cell culture and cloning, Lukas Huber for providing the vectors for shRNA cloning, Anja Wolf for AAV generation, Anna Falk and Gillian Morrison for neural and endoderm differentiation, and Gabi Lederer (Cytogenetic Department, TUM) for karyotyping. This work was supported by grants from: the European Research Council, ERC 261053 (to K.-L.L.); the German Research Foundation, Research Unit 923, Mo 2217/1-1 (to A.M.) and La 1238 3-1/4-1 (to K.-L.L.), and Transregio Research Unit 152 (to A.M., K-L.L.); the German Ministry for Education and Research, 01 GN 0826 (to K.-L.L and A.M.) and the UK Medical Research Council (to A.S.). H.A. was funded by a Wellcome Trust Clinical Research fellowship. A.S. is a Medical Research Council Professor.

## Author contributions

TD, JH, EIP, and HA designed and performed most of the experiments and analyzed data. DZ, LI, IM, SL, TB, CB, TZ, and RH performed experiments. GS performed the bioinformatics analyses. EM, AG, EG, and TM performed exome and ChIP sequencing and data analysis. DS and RJD designed experiments, analyzed data, and provided conceptual advice. SK and AAG provided human patient biopsies. CK, GC, RG, and AGS provided financial support and conceptual advice. K-LL and AM conceived and supervised this study and provided financial support. AM wrote the paper. All authors commented on and edited the manuscript.

## Conflict of interest

The authors declare that they have no conflict of interest.

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
