## [Review Process File · The EMBO Journal]

Interplay of cell-cell contacts and RhoA/MRTF-A signaling regulates cardiomyocyte identity

Tatjana Dorn, Jessica Kornherr, Elvira I. Parrotta, Dorota Zawada, Harold Ayetey, Gianluca Santamaria, Laura Iop, Elisa Mastantuono, Daniel Sinnecker, Alexander Goedel, Ralf J. Dirschinger, Ilaria My, Svenja Laue, Tarik Bozoglu, Christian Baarlink, Tilman Ziegler, Elisabeth Graf, Rabea Hinkel, Giovanni Cuda, Stefan Kääh, Andrew A. Grace, Robert Grosse, Christian Kupatt, Thomas Meitinger, Austin G. Smith, Karl-Ludwig Laugwitz, Alessandra Moretti

Review timeline:

Submission date:	1 September 2017
Editorial Decision:	9 October 2017
Additional correspondence:	9 December 2017
Additional correspondence:	11 December 2017
Revision received:	28 February 2018
Editorial Decision:	26 March 2018
Revision received:	29 March 2018
Accepted:	4 April 2018

Editors: Ieva Gailite, Deniz Senyilmaz Tiebe

Transaction Report:

1st Editorial Decision

9 October 2017

Thank you for submitting your manuscript for consideration by the EMBO Journal. We have now finally received all three reports on your manuscript, which I am including below for your information.

As you can see from the comments, all three referees express interest in the proposed mechanism of cardiomyocyte adipogenic transdifferentiation. However, they also raise substantive concerns with the analysis that would need to be addressed before they can support publication here. Based on the overall interest expressed in the reports I would like to invite you to submit a revised version of your manuscript in which you address the comments of all three referees. I would ask you to focus in particular on the following points:

- Investigate the ability of mature cardiomyocytes to differentiate into adipocytes (Reviewer #1, point 1, and Reviewer #3, point 2).
- Please provide better validation of the Isl1+/Wt1+ cell fate by lineage tracing (Reviewer #3, point 1)
- Provide further validation of the brown adipocyte identity of the cardiomyocyte-derived cells (Reviewer #1, point 2)
- Please provide analysis of the effect of Wt1 loss of function on adipocyte differentiation (Reviewer #2, point 1)
- Provide data on the interaction of PPAR γ with Wt1 during adipocyte differentiation (Reviewer #2, point 2).

I should add that it is The EMBO Journal policy to allow only a single major round of revision and that it is therefore important to resolve the main concerns raised at this stage. Since extensive additional work would be needed to fulfill all the referee requests, please contact me if you would like to discuss the feasibility of any of the experiments for the revision, or if you were to choose not to undergo an extensive revision here and rather pursue a submission at an alternative venue.

We generally allow three months as standard revision time, although an extension to up to six months is possible in the case of extensive revisions. Please contact us in advance if you would need an additional extension. As a matter of policy, competing manuscripts published during this period will not negatively impact on our assessment of the conceptual advance presented by your study. However, we request that you contact the editor as soon as possible upon publication of any related work to discuss how to proceed.

Please feel free to contact me if have any further questions regarding the revision. Thank you for the opportunity to consider your work for publication. I look forward to your revision.

REFEREE COMMENTS

Referee #1:

Moretti, Laugwitz and colleagues examined cardiomyocyte-to-adipocyte conversion in the context of arrhythmogenic right ventricular cardiomyopathy. Utilizing *in vivo* lineage tracing coupled with *in vitro* cellular manipulations, the authors demonstrate that cardiac adipose and a subset of cardiac muscle cells arise from a common or similar set of precursors that express *Isl1* and *Wt1* during development. The paper also provides evidence that mechanosensory pathways are required to maintain cardiomyocyte vs. adipocyte identity. Overall, I find the studies to be very compelling and rigorously executed. I think this is a very interesting and complete paper which is suitable for publication in EMBOEmbo J. I have only two minor comments for the authors to consider.

1. The studies indicate that cardiomyocytes can transition to an adipocyte-like fate under certain conditions and that certain precursors have the capacity to differentiate into both lineages when exposed to certain stimuli. However, there is no direct evidence that mature cardiomyocytes transform into adipocytes *in vivo*. While this conclusion seems likely, it is also possible that a population of resident precursor-like cells are an important source of adipocytes *in vivo*. To definitively address this, one would need to lineage trace the fate of mature cardiomyocytes *in vivo* in response to the PKP2 mutation. A discussion around this point would be appropriate.
2. The authors argue that oil-red-o accumulation and brown-adipocyte morphology represents a fundamental lineage switch. Additional studies would be needed to define these cells as bona fide brown fat cells, including respiratory capacity, glucose uptake, insulin responsiveness, etc. The glycerol release assays are not that convincing since authentic brown fat cells have higher activity in this assay. It would also be important to compare the level of brown gene expression in the cardiomyocyte-derived cells with "true" brown fat cells ie. from rodent interscapular brown fat. Without additional characterization, it would be more prudent to refer to the identified cells as: "multilocular brown adipocyte-like cells" or similar.

Referee #2:

This study by Dorn and colleagues addresses the underlying mechanism of cardiomyocyte (CM)-adipocyte conversion in arrhythmogenic right ventricular cardiomyopathy (ARVC). The authors describe the lineage conversion in human ARVC patient iPSC-CMs (harbouring a PKP2 mutation) and carried-out fate mapping studies in mice to reveal a subset of *Isl-1+/Wt1+* progenitors that contribute either CMs or AV groove adipocytes within the AV groove and demonstrated that these progenitors can give rise to both CMs and fat cells *in vitro*. They reveal a switch in cytoplasmic to nuclear WT-1 expression in CMs undergoing adipocyte conversion, which was insufficient to drive the process but sustained the conversion as induced by PPARgamma. Comparing whole transcriptomes of PKP2 versus wild type CMs they identified adipogenic pathway enrichment and

an activated histone signature (increased H3K4me and H3K27ac occupancy) at the PPARgamma locus. Notably the transcriptomic data revealed dysregulation of genes associated with ROCK pathway; in support of this RhoA was reduced in PKP2-mutated CMs coincident with increased G-actin and cytosolic actin disruption during CM-adipocyte conversion. Based on a previous study implicating RhoA- regulation of MRTF-A in adipocyte differentiation, the authors then tested a role for MRTFA in CM conversion and noted over-expression of MRTFA could rescue the PKP2-mutant phenotype, reduce PPARgamma activity and block CM to adipocyte conversion. The role for ROCK signalling in triggering CM-adipocyte conversion was further supported by chemical inhibition of RhoA and the analysis of ARVC patient iPSC lines harbouring MYH10 mutations, whereby MYH10 has previously been implicated in actin dynamics and RhoA-GTP accumulation at the intercalated disc in CMs. Prolonged treatment of WT CMs with the RhoA inhibitor Y27632 resulted in a modest appearance of adipocyte-like cells and a subset of MYH10-mutant cells exhibited nuclear translocation of WT1, activation of PPARgamma and ultimate conversion into adipocytes. This study is comprehensive and the data in general is high quality. It builds significantly on previous studies in a number of key areas: ARVC modelling in patient-derived iPSCs (eg. Caspi et al., 2013); fate mapping studies that revealed a second heart field contribution to adipocytes in a Dsp- deficient mice and human ARVC samples (Lombardi et al. 2009) and the implication of ROCK signalling (Nobusue et al., 2014) and MRTFA (McDonald et al. 2015) in adipogenesis per se. Importantly it provides novel functional insight into the role of altered mechanosensing of cell-cell contact and convergence onto a RhoA/MRTF/SRF regulatory pathway for CM lineage conversion and adipogenesis. Thus, this study adds significantly to our understanding of the maintenance of human CM fate as well as the underlying pathology of ARVC.

There are a few issues that should be addressed:

1. The authors tested whether WT1 might be sufficient to induce CM to adipocyte conversion but ought to investigate a corresponding loss-of-function model. For example, Wt1-GFPCre mice, to determine whether Wt1 might be necessary for adipocytes formation within the AV groove in mice or whether in Wt1-targeted hiPSC-CMs PPARgamma-induced conversion is reduced(?).
2. The authors do not currently provide any data to support the cooperative role of Wt1 and PPARgamma- as proposed on page 10. Do WT1 and PPARgamma physically interact by co-IP?- some biochemical evidence of cooperative function is important here alongside the suggested genetic experiment under 1) above.
3. The authors should test whether the limited ability of the RhoA inhibitor Y27632 to induce adipocyte-like cells, can be augmented if the inhibitor is targeted to YFP/Isl-1+ mESC derived CMs (as per the model presented in Figure 3). This would be highly informative regarding the role of RhoA, but also in addressing the speculation (on pages 14 and 16) to whether the fate-mapped progenitors represent a subset of CMs that are more prone to adipocyte conversion based on their developmental origin/epigenetic status.

Minor points:

1. The "lipogenic milieu" ought to be included in the main text on page 4- as this would be informative up-front regarding the adipogenic signals.
2. Page 8- "which are its extreme paucity" is presumably a typo(?), and needs correcting.
3. Figure 2- immunostaining labels: mG, DNA etc, are not visible against the background in most of the panels.
4. Page 11- why 3 months of cardiac differentiation for the genome expression profiling? There needs some justification/rationale inserting here.

Referee #3:

The manuscript by Dorn et al., describes direct transdifferentiation of iPS derived CM into adipocytes, positing a novel source of fatty tissue deposition in ARVC. The authors find that Isl1 and WT1 double positive cells differentiate into CM that can subsequently generate adipocytes, particularly upon culture in adipogenic medium. The authors go on to show that iPS derived CM from Pkp2 mutant or Myh10 mutant patients also can generate brown fat. The data are intriguing and potentially point towards a novel cellular source of adipocytes in response to alterations in Rho-dependent SRF/MRTF signaling upon mutation or loss of PKP2 at the desmosome; however, several fundamental issues reduce enthusiasm. Importantly, it is difficult to conceive an adult CM in an ARVC patient behaving as the iPS derived cells do in vitro, which may be predisposed to

artifacts of culture and immature differentiation state.

Major points;

1. The authors argue that Isl1/Wt1 double positive progenitor-derived CM are sensitive to transdifferentiating into adipocytes as a novel cellular source of fat deposition in ARVC. Lineage tracing of Isl1 and Wt1 + cells have previously been performed that indicate pro-epicardial origin and CM fate (For Isl1). WT1 is not thought to contribute to the CM lineage in general, therefore a better description of the cell populations in vivo is necessary. The authors focus on the 1 potential CM that is derived from WT1CreERT2 in Fig. 2B, but it is not clear how many are derived from this population and whether this is right or left ventricle. For example, without presenting the data, the authors mention that only 50-70 cells per embryo are Isl1/WT1 double positive. Do these cells differentiate into CM? Do they remain progenitors? Do they reach adequate numbers to be functionally relevant in disease? Tracing this population in vivo using a method such as Isl1-Cre/Wt1-Dre in a double reporter background (Rosa-Ai66) would be required to adequately evaluate the fate of this population in vivo. This is especially true since the majority of the paper evaluates the propensity of iPS derived Isl1+ CM to transform into adipocytes directly. Indeed, better characterization of the WT1-CreERT2;Rosa-mTmG mouse upon dosing with TMX at E7.5 (very early) is warranted. The WT1-CreGFP line is reported to have an early and transient ubiquitous expression domain just prior to E9.5. Since the Wt1-CreERT2 line is at the identical locus, one might worry that an early ectopic pattern may be present in this line as well, coincident with E7.5 TMX dose and leading to ectopic labeling.
2. Using iPS derived CM to suggest a novel cellular source of adipogenesis in ARVC is artifact prone. This immature CM population might be susceptible to differentiate into other cell types in vitro. Furthermore, ARVC is thought to become evident primarily in the adult following stress such as exercise. iPS-derived CM is not an adequate model of mature adult CM undergoing mechanical stress. Convincing evidence that CM contribute directly to the adipocyte population in vivo is lacking. It is difficult to imagine mature adult CM transdifferentiating into adipocytes in ARVC.
3. The population of cells supposed to be involved in vivo (Isl1 progenitors that no longer express Isl1 but do express Wt1) are not the same as the cells used in all in vitro experiments (cells that express both Isl1 and Wt1). The correlation between in vivo experiments and in vitro studies is therefore not strong. WT1 staining in control and ARVC patient hearts is not convincing and does not agree with the literature.
4. Page 9 states that they see WT1 expression in CMs of ARVC patients. However this contradicts both lineage tracing and immunostaining studies, and also contradicts their reference (Ambu et al., 2015) that states WT1 is observed in fetal CM (not adult).
5. The authors interpret reduced PKP2 levels observed as "compromised cell-cell adhesion" throughout the manuscript without evidence this is the case. Evaluating physical cell associations would be required. Loss of cellular contacts typically leads to MRTF accumulation in the nucleus, a contradictory finding that is not commented on. How does MRTF localization and PPAR γ expression respond to physical or chemical disruption of cell-cell contacts?

Minor points:

6. Quantification of the % / number of WT1 / ISL1 positive cells, both by lineage tracing (Fig. 2) and by immunofluorescence (Fig. 3F) is required.

Additional correspondence (author)

9 December 2017

Thank you very much for your kind letter on the 9th of October 2017 inviting us to submit a revised manuscript on our work addressing the underlying mechanism of cardiomyocyte-to-adipocyte direct conversion in arrhythmogenic right ventricular cardiomyopathy. We are very thankful to you and the reviewers for the critical and constructive input during the review of our manuscript. In the past 2 months, we have performed several new experiments in order to address your and the referees' comments. These include:

- 1) detailed characterization of the ED8 Isl1/Wt1 lineage descendants (amount of cells and their specific localization) in the developing heart and proepicardial organ at ED9.5 as well as in the

myocardium and fat depot of adult mice, using *Isl1MerCreMer/+;R26mTmG/+* and *Wt1CreERT2/+;R26mTmG/+* animals upon dosing with tamoxifen at ED7.5. This new extended analysis clearly demonstrate the existence, at both developmental time points, of a cellular overlap between the two lineage tracings in terms of cell types and their organ localization, strongly suggesting that these specific cells derive from a common progenitor that express both *Isl1* and *Wt1* at the time of tamoxifen injection. Moreover, we found that labelling of *Wt1CreERT2/+;R26mTmG/+* ED9.5 embryos was restricted to the proepicardial organ/epicardium and the urogenital tract, both *Wt1*-lineage derivatives, indicating the specificity of our marking by ED7.5 tamoxifen injection (Referee #3, point 1).

Since, to our knowledge, no *Dre* reporter lines have been yet generated for either the *Isl1* and/or the *Wt1* lineage, the lineage tracing with a combined *Cre/Dre;Rosa-Ai66* background suggested by Referee #3 is practicably very time-consuming and cannot be achievable even with a revision extension of 6 months. We agree that this approach would be the ideal one for quantifying the exact contribution of the *Isl1+/Wt1+* lineage to the various heart structures *in vivo*. However, we believe that our new extended analyses of the ED9.5 embryos and adult hearts of *Isl1MerCreMer/+;R26mTmG/+* and *Wt1CreERT2/+;R26mTmG/+* mice, in combination with our previous *in vitro* clonal experiments using mouse ESC-derived *Isl1+/Wt1+* progenitors, provide very strong evidence that a cardiac precursor population expressing *Isl1* and *Wt1* exist, which is capable of generating both cardiomyocytes and adipocytes *in vivo* and *in vitro*.

2) Co-IP experiments showing that indeed *Wt1* and *PPARg* can physically interact, supporting the cooperative role of *Wt1* and *PPARg* in driving the transcriptional changes leading to the cardiomyocyte-adipocyte switch. These results have been so far obtained in HEK293 cells, and still need to be validated during cardiomyocyte-adipocyte transdifferentiation.

3) further validation of the brown adipocyte identity of the converted cardiomyocytes by i) comparison of the level of brown fat gene expression in our cardiomyocyte-derived adipocytes with those from mouse native interscapular brown fat and ii) functional assessment of oxygen consumption and respiratory capacity.

4) Validation of shRNAs for efficient *WT1* silencing. We are now ready to investigate the effect of *WT1* loss-of-function in the cardiomyocyte-to-adipocyte conversion process of WT cardiomyocytes after *PPARg* overexpression.

The most time-consuming experiment, which is still work in progress, is the investigation of the ability of mature cardiomyocytes to convert into adipocytes. We have preliminary evidence that adult cardiomyocytes isolated from the human heart can acquire an adipocyte-like phenotype upon forced expression of *PPARg* *in vitro*. We are now in the process to evaluate whether this can occur also *in vivo* by tracing the fate of adult mature cardiomyocytes after *WT1* and *PPARg* overexpression (single and in combination) in *•MHCCre/+; R26mTmG/+* mice. Selective transgene expression in cardiomyocytes will be achieved using recombinant adeno-associated viral vectors serotype 9 (AAV9), which have a preferential tropism for muscle cells and are not immunogenic. This experiment will not only reinforce our *in vitro* results demonstrating that *PPARg* is the molecular trigger of the adipogenic conversion, but also allow us to evaluate whether *WT1* is essential for the cardiomyocyte-adipocyte switch. This is of particular interest, in our opinion, since *Wt1*, differently from human, is not expressed in mouse cardiomyocytes, which may reflect the relative low amount of cardiac fat in mouse compared to human and the fact that all reported ARVC mouse models lacking specific desmosomal genes, although they develop arrhythmias, show only very limited adipogenic phenotypes. This latter aspect has hampered a definitive identification of the cellular origin of excess adipocytes in ARVC pathology. However, conditional ablation of the desmosomal protein desmoplakin in the cardiac lineage using reporter mice allowing either deletion at the stage of adult cardiomyocyte (*•MHCCre/+;Dspflx/flx; R26EYFP/+* mice) or earlier in cardiac progenitors (*Nkx••Cre/+;Dspflx/flx; R26EYFP/+* and *Mef2cCre/+;Dspflx/flx; R26EYFP/+*) has provided first evidence that the "extra cardiac adipocytes" in this animals can originate from mature cardiomyocytes (Lombardi et al, *Circ Res* 2009). Moreover, the finding that desmoplakin deletion already at the progenitor stage resulted in a higher number of reporter-labelled adipocytes corroborates our hypothesis that a "priming" for the cardiomyocyte-adipocyte switch occurs and this is likely at the level of a progenitor.

We hope that our planned experiments with WT1 and PPAR γ cardiac overexpression *in vivo* will provide novel new insights on the mechanism of cardiomyocytes-to-adipocyte conversion in ARVC. In order to complete our revision we kindly ask for an extension of 3 more months. However, we hope to be able to resubmit our work by the end of February 2018.

In advance, I thank you for your time and consideration.

Additional correspondence (editor)

11 December 2017

Thank you for the update on the revision progress. I am glad to hear that you are/will be able to address most of the core issues that were pointed out by the reviewers. I of course understand that there are cases of experimental and time limitations, which you are then welcome to discuss in the point-by-point response. Due to the extensive nature of the revision, I am happy to extend the resubmission deadline to February 28. I am looking forward to receiving the revised version!

1st Revision - authors' response

28 February 2018

POINT-BY-POINT RESPONSE TO REFEREES

We thank the reviewers for their interest in our work, constructive criticisms and instructive comments. We have addressed each of the major issues they raised by carrying out an extensive set of new experiments, as well as *via* editorial revision. We believe these clarify key issues highlighted in their review. These changes have been incorporated into the revised version of the manuscript and Supplementary Appendix. The new manuscript text and Figures, as well as a point-by-point rebuttal are provided for the referees and the editor.

Summary of the major changes: at the suggestion of Reviewer #1, we further validated the brown adipocyte identity of the converted cardiomyocytes at both molecular and functional levels by i) comparing of the expression of brown fat genes in our cardiomyocyte-derived adipocytes with those from mouse native interscapular brown fat and human brown/beige adipocytes obtained by directed differentiation of hiPSCs, and ii) functional assessment of oxygen consumption and respiratory capacity as hallmark of mitochondrial function. These analyses have resulted in a new panel in the main Figure 1 (panel I) and two new panels in the Expanded View Figure EV1 (panels E and F) and indicate similar characteristics of the converted cells to brown/beige adipocytes. Moreover, as recommended by Reviewer #2, a new set of experiments addressed the effects of WT1 loss-of-function on the cardiomyocyte-to-adipocyte conversion and demonstrated protein-protein interaction of WT1 and PPAR γ during the conversion process. This resulted in a revised Figure 4 (panels F, H and I) and a new Supplementary Figure (Appendix Fig S6 of the revised manuscript). In addition, as requested by Reviewer #3, we have extended the analysis of the E8 Isl1/Wt1 lineage descendants in the developing heart and proepicardial organ at E9.5 as well as in the myocardium and cardiac fat depot of adult mice. This resulted in two new Expanded View Figures (Figures EV2 and EV3) of the revised manuscript. Finally, at recommendation of Reviewers #1 and #3, a new Figure EV5 provides now direct evidence that also mouse adult cardiomyocytes can convert into adipocytes *in vivo* when forced to overexpress PPAR γ and WT1, further confirming PPAR γ as molecular trigger of the adipogenic conversion and the requirement of WT1 in this process. Lastly, we have revised the Introduction section to highlight published lineage tracing studies in ARVC mouse models providing evidence that pathological fat in this disease condition can arise from mature cardiomyocytes and this process is “primed” early during cardiogenesis.

We believe that our revised manuscript addresses all of the reviewers' concerns as summarized in the point-by-point response below.

Referee #1:

Moretti, Laugwitz and colleagues examined cardiomyocyte-to-adipocyte conversion in the context of arrhythmogenic right ventricular cardiomyopathy. Utilizing in vivo lineage tracing coupled with in vitro cellular manipulations, the authors demonstrate that cardiac adipose and a subset of cardiac muscle cells arise from a common or similar set of precursors that express Islet1 and Wt1 during development. The paper also provides evidence that mechanosensory pathways are required to maintain cardiomyocyte vs. adipocyte identity. Overall, I find the studies to be very compelling and rigorously executed. I think this is a very interesting and complete paper which is suitable for publication in EMBO J. I have only two minor comments for the authors to consider.

We thank the referee for the largely positive comments, constructive criticism, and insights that helped us to improve the manuscript.

The specific responses to each of the points are noted below:

1. The studies indicate that cardiomyocytes can transition to an adipocyte-like fate under certain conditions and that certain precursors have the capacity to differentiate into both lineages when exposed to certain stimuli. However, there is no direct evidence that mature cardiomyocytes transform into adipocytes in vivo. While this conclusion seems likely, it is also possible that a population of resident precursor-like cells are an important source of adipocytes in vivo. To definitively address this, one would need to lineage trace the fate of mature cardiomyocytes in vivo in response to the PKP2 mutation. A discussion around this point would be appropriate.

We thank the reviewer for this important suggestion. Indeed, previous fate mapping studies from the Ali Marian group have provided first evidence in mouse that mature adult cardiomyocytes can convert into adipocytes in vivo in the setting of human autosomal dominant ARVC, e.g. in presence of heterozygous deficiency of the desmosomal protein desmoplakin (Lombardi et al., Circ Res 2009). Using aMHC^{Cre/+};Dsp^{flx/flx};R26^{EYFP/+} mice, which allow simultaneous ablation of desmoplakin in adult cardiomyocytes and their lineage tracing, the authors detected few adipocytes labelled by the EYFP reporter. Interestingly, when desmoplakin was ablated earlier at the stage of cardiac progenitors using either Mef2C^{Cre/+};Dsp^{flx/flx};R26^{EYFP/+} or Nkx2.5^{Cre/+};Dsp^{flx/flx};R26^{EYFP/+} mice, not only the overall number of pathological “excess of adipocytes” was increased but also the percentage of adipocytes expressing EYFP. Since in the latter mice cardiomyocytes are irreversibly marked by EYFP, this could suggest a higher efficiency of cardiomyocyte-to-adipocyte conversion when ARVC-causing mutations “act” early during cardiogenesis. This is in agreement with our findings in patient iPSC-derived cardiomyocytes showing that “priming” for the cardiomyocyte-adipocyte switch occurs and this is likely at the level of a progenitor cell. We have now highlighted the study of Lombardi et al. in the Introduction section of the revised manuscript on page 3, second paragraph.

To further validate the capability of mature cardiomyocytes to transdifferentiate into adipocytes *in vivo* under the conditions experienced by ARVC myocytes (high level of PPAR γ and nuclear WT1), we overexpressed NLS-WT1 and PPAR γ (alone and in combination) in adult myocytes of Myh6^{Cre/+};R26^{mTmG/+} mice using adeno-associated viruses serotype 9 (AAV9-NLS-WT1 and AAV9-PPAR γ), which display high tropism to cardiac and skeletal muscles. Tail vein injection of 2.5×10^{12} virus particles resulted in 30-40 infected mG⁺ cells per heart slice. Interestingly, we observed emergence of few intramyocardial lipid-filled/Plin⁺ adipocytes labelled by mG exclusively in hearts of animals that received both viruses (2-3 Plin⁺/mG⁺ cells per heart slice at 5 weeks post-infection). Since WT1, differently from human, is not expressed in mouse cardiomyocytes, these results hint to a requirement of WT1 for the conversion process. This is also confirmed by our additional new data demonstrating that silencing of WT1 in human iPSC-CMs hampers their PPAR γ -mediated adipocyte conversion (revised Figure 4, panels F and H, and new Appendix Figure S6). Moreover, they may explain the relative low amount of cardiac fat in mouse compared to human and the fact that all reported ARVC mouse models lacking specific desmosomal genes, although they develop arrhythmias, show only very limited adipogenic phenotypes.

These new data are now described in the Result section on page 10, second paragraph, and page 11, first paragraph, and illustrated in the new Expanded View Figure EV5 of the revised manuscript.

2. The authors argue that oil-red-o accumulation and brown-adipocyte morphology represents a fundamental lineage switch. Additional studies would be needed to define these cells as bona fide brown fat cells, including respiratory capacity, glucose uptake, insulin responsiveness, etc. The glycerol release assays are not that convincing since authentic brown fat cells have higher activity in this assay. It would also be important to compare the level of brown gene expression in the cardiomyocyte-derived cells with "true" brown fat cells ie. from rodent interscapular brown fat. Without additional characterization, it would be more prudent to refer to the identified cells as: "multilocular brown adipocyte-like cells" or similar.

As suggested, we have now further validated the brown adipocyte identity of the converted cardiomyocytes at both molecular and functional levels by i) comparing the expression of brown fat genes in our cardiomyocyte-derived adipocytes with those from mouse native interscapular brown fat and human brown/beige adipocytes obtained by directed differentiation of hiPSCs, ii) functional assessment of oxygen consumption rate (OCR) as hallmark of mitochondrial function, and iii) activation of the thermogenic gene program under β -adrenergic stimulation. When compared to mouse native interscapular (ISC) brown fat, expression of some brown fat genes such as *CIDEA*, *UCP1*, and *ELOVL3* was lower in our PKP2-mutant cardiomyocyte-derived adipocyte-like cells, while for other genes such as *PRDM16* no differences were observed, as illustrated here for the reviewer only. The numbers shown are normalization to *GAPDH* and relative to ISC (\pm SEM from 3 experiments):

These differential expressions are likely attributable to the facts that i) quantitative RT-PCR primers used were different between human and mouse samples, thus hindering an accurate quantitative comparison in gene expression levels, and most importantly ii) brown fat deposits in adult humans seem to be heterogeneous and share more molecular features with beige than brown adipocytes described in rodents (Cohen & Spiegelman, Diabetes 2015). Therefore, we extended our analysis to human brown/beige adipocytes obtained from hiPSCs, which have been recently shown to resemble quite well native human brown/beige fat (Ahfeldt et al., Nat Cell Biol 2012; Guenantin et al., Diabetes 2017). Directed differentiation of hiPSCs to brown/beige adipocytes was performed as recently described by Guenantin et al. Differentiated adipocytes showed similar level of expression of *PPARG*, *UCP1*, and *TMEM26*, among others, as reported by Guenantin et al. Most importantly, expression of these transcripts was comparable between hiPSC-derived adipocytes and our PKP2-mutant cardiomyocyte-derived adipocyte-like cells, as well as activation of the thermogenic gene program upon treatment with the cAMP analog 8Br-cAMP for 48h, suggesting a “true” conversion of our cells towards the brown/beige lineage. These new data have been included in the new Expanded View Figure EV1 (panels E and F) of the revised manuscript and described in the Result section on page 5, second paragraph, and page 6, first paragraph.

Moreover, assessments of oxygen consumption rate (OCR) using an extracellular flux analyser also revealed that the mitochondrial function of the mutated “converted” cells was very similar to that of human iPSC-derived brown/beige adipocytes and differed from that of CMs. These new data are now described in the Result section on page 6, first paragraph, and presented in the revised Figure 1 (panel H) of the revised manuscript.

For clarity, we now refer to a “brown/beige” fat identity throughout the revised manuscript.

Referee #2:

This study by Dorn and colleagues addresses the underlying mechanism of cardiomyocyte (CM)-adipocyte conversion in arrhythmogenic right ventricular cardiomyopathy (ARVC). The authors describe the lineage conversion in human ARVC patient iPSC-CMs (harbouring a PKP2 mutation) and carried-out fate mapping studies in mice to reveal a subset of Isl-1+/Wt1+ progenitors that contribute either CMs or AV groove adipocytes within the AV groove and demonstrated that these progenitors can give rise to both CMs and fat cells in vitro. They reveal a switch in cytoplasmic to nuclear WT-1 expression in CMs undergoing adipocyte conversion, which was insufficient to drive the process but sustained the conversion as induced by PPARgamma. Comparing whole transcriptomes of PKP2 versus wild type CMs they identified adipogenic pathway enrichment and an activated histone signature (increased H3K4me and H3K27ac occupancy) at the PPARgamma locus. Notably the transcriptomic data revealed dysregulation of genes associated with ROCK pathway; in support of this RhoA was reduced in PKP2-mutated CMs coincident with increased G-actin and cytosolic actin disruption during CM-adipocyte conversion. Based on a previous study implicating RhoA- regulation of MRTFA in adipocyte differentiation, the authors then tested a role for MRTFA in CM conversion and noted over-expression of MRTFA could rescue the PKP2-mutant phenotype, reduce PPARgamma activity and block CM to adipocyte conversion. The role for ROCK signalling in triggering CM-adipocyte conversion was further supported by chemical inhibition of RhoA and the analysis of ARVC patient iPSC lines harbouring MYH10 mutations, whereby MYH10 has previously been implicated in actin dynamics and RhoA-GTP accumulation at the intercalated disc in CMs. Prolonged treatment of WT CMs with the RhoA inhibitor Y27632 resulted in a modest appearance of adipocyte-like cells and a subset of MYH10-mutant cells exhibited nuclear translocation of WT1, activation of PPARgamma and ultimate conversion into adipocytes. This study is comprehensive and the data in general is high quality. It builds significantly on previous studies in a number of key areas: ARVC modelling in patient-derived iPSCs (eg. Caspi et al., 2013); fate mapping studies that revealed a second heart field contribution to adipocytes in a Dsp- deficient mice and human ARVC samples (Lombardi et al. 2009) and the implication of ROCK signalling (Nobusue et al., 2014) and MRTFA (McDonald et al. 2015) in adipogenesis per se. Importantly it provides novel functional insight into the role of altered mechanosensing of cell-cell contact and convergence onto a RhoA/MRTF/SRF regulatory pathway for CM lineage conversion and adipogenesis. Thus, this study adds significantly to our understanding of the maintenance of human CM fate as well as the underlying pathology of ARVC. There are a few issues that should be addressed:

We thank the referee for the appreciation of our work and the efforts and critical input during the review of our manuscript. The specific responses to each of the points are noted below:

1. The authors tested whether WT1 might be sufficient to induce CM to adipocyte conversion but ought to investigate a corresponding loss-of-function model. For example, Wt1-GFP^{Cre} mice, to determine whether Wt1 might be necessary for adipocytes formation within the AV groove in mice or whether in Wt1-targeted hiPSC-CMs PPARgamma-induced conversion is reduced(?)

Since both global and epicardial-specific knockout of Wt1 causes embryonic lethality (Kreidberg et al. Cell 1993; Martinez-Estrada et al. Nature Genetics 2010), hampering the investigation of AV groove fat that is first visible at 2 weeks of postnatal life, we follow the second referee's suggestion and tested whether knock-down of endogenous WT1 in PPARg-overexpressing wt CMs would negatively affect their conversion. Indeed, by day 21, cultures expressing shRNA targeting WT1 showed a striking, dramatic reduction in lipid-filled adipocyte-like cells and levels of adipocytic genes, while scrambled shRNA had no effects. These new data, which reinforce a cooperative role of WT1 and PPARg in driving human cardiomyocyte-to-adipocyte conversion and even point to a requirement of WT1 for PPARg-mediated cardiomyocyte-to-adipocyte switch, have been incorporated in panels F and H of the revised Figure 4 and in the new Appendix Fig S6 and are described in the Result section on page 10, second paragraph.

2. The authors do not currently provide any data to support the cooperative role of *Wt1* and *PPARgamma*- as proposed on page 10. Do *WT1* and *PPARgamma* physically interact by co-IP?- some biochemical evidence of cooperative function is important here alongside the suggested genetic experiment under 1) above.

We thank the referee for insightful comments and constructive remarks that encouraged us to further evaluate the role of *WT1* in supporting *PPARg*-induced adipocytic switch and to analyse potential protein-protein interaction between *WT1* and *PPARg*. Following the referee's suggestion, we examined this hypothesis by overexpressing HA-tagged *PPARg* and NLS-*WT1* in wt CMs and performing co-IP experiments in nuclear lysates during cardiomyocyte-to-adipocyte conversion. Indeed, *WT1* was detected in the HA immunoprecipitation complex and *PPARg* was also co-immunoprecipitated with *WT1*, confirming their physical interaction. These data are now illustrated in the revised Figure 4I and explained in the Result section on page 10, second paragraph.

3. The authors should test whether the limited ability of the *RhoA* inhibitor Y27632 to induce adipocyte-like cells, can be augmented if the inhibitor is targeted to *YFP/Isl-1+* mESC derived CMs (as per the model presented in Figure 3). This would be highly informative regarding the role of *RhoA*, but also in addressing the speculation (on pages 14 and 16) to whether the fate-mapped progenitors represent a subset of CMs that are more prone to adipocyte conversion based on their developmental origin/epigenetic status.

We thank the referee for this valuable comment. Indeed, we have data, which were not included in the original version of the manuscript, suggesting that CMs differentiated from mESC-derived *Isl1⁺/Wt1⁺* clones are more "prone" to adipogenesis than CMs obtained from *Isl1⁺/Wt1⁻* clones when subjected to *RhoA*/*ROCK* inhibition. Specifically, after induction of CM differentiation in myogenic medium we had subsequently treated some cultures with adipogenic medium and observed accumulation of lipid droplets merely in CMs from *Isl1⁺/Wt1⁺* clones, suggesting a higher plasticity of these differentiating CMs towards the adipocytic fate. Since adipogenic medium contains IBMX and dexamethasone, which inhibit *RhoA*/*ROCK* activity by activating protein kinase A (Petersen *et al*, Mol Cell Biol 2008), it is likely that further *ROCK* inhibition by Y-27632 would potentiate adipogenesis in these cells.

We have now presented these results in the revised manuscript on page 8, first paragraph, and in the new Appendix Fig S4.

Minor points:

1. The "lipogenic milieu" ought to be included in the main text on page 4- as this would be informative up-front regarding the adipogenic signals.

Due to word limitation, we needed to drastically shorten our manuscript. Therefore, we describe the composition of the lipogenic medium only in the Appendix Methods, and refer the readers to this section on page 4 of the main manuscript.

2. Page 8- "which are its extreme paucity" is presumably a typo(?), and needs correcting.

We thank the referee for his note. However, during the editorial revision of the manuscript this sentence has been removed.

3. Figure 2- immunostaining labels: mG, DNA etc, are not visible against the background in most of the panels.

We thank the referee for this remark. We have now improved the labelling.

4. Page 11- why 3 months of cardiac differentiation for the genome expression profiling? There needs some justification/rationale inserting here.

Time of culture has been proven to improve maturation of iPSC-derived cardiomyocytes. At 2-3 months differentiation, cells have reached a sufficient level of functional and molecular "maturity"

and only little changes occur with further culture. Thus, we used 2-3-month-old cells in all our experiments.

For better clarity, we have now removed the “3 months of cardiac differentiation” on page 11.

However, the age of the cells used in the experiments is specified in the Appendix Methods section under “Generation, culture and *in vitro* differentiation of human iPSCs”.

Referee #3:

The manuscript by Dorn et al., describes direct transdifferentiation of iPS derived CM into adipocytes, positing a novel source of fatty tissue deposition in ARVC. The authors find that Isl1 and WT1 double positive cells differentiate into CM that can subsequently generate adipocytes, particularly upon culture in adipogenic medium. The authors go on to show that iPS derived CM from Pkp2 mutant or Myh10 mutant patients also can generate brown fat. The data are intriguing and potentially point towards a novel cellular source of adipocytes in response to alterations in Rho-dependent SRF/MRTF signaling upon mutation or loss of PKP2 at the desmosome; however, several fundamental issues reduce enthusiasm. Importantly, it is difficult to conceive an adult CM in an ARVC patient behaving as the iPS derived cells do in vitro, which may be predisposed to artifacts of culture and immature differentiation state.

We thank the referee for the largely positive comments, constructive criticism, and insights that helped us to improve the manuscript. The specific responses to each of the points are noted below:

Major points;

1. The authors argue that Isl1/Wt1 double positive progenitor-derived CM are sensitive to transdifferentiating into adipocytes as a novel cellular source of fat deposition in ARVC. Lineage tracing of Isl1 and Wt1 + cells have previously been performed that indicate proepicardial origin and CM fate (For Isl1). WT1 is not thought to contribute to the CM lineage in general, therefore a better description of the cell populations in vivo is necessary. The authors focus on the 1 potential CM that is derived from WT1CreERT2 in Fig. 2B, but it is not clear how many are derived from this population and whether this is right or left ventricle. For example, without presenting the data, the authors mention that only 50-70 cells per embryo are Isl1/WT1 double positive. Do these cells differentiate into CM? Do they remain progenitors? Do they reach adequate numbers to be functionally relevant in disease? Tracing this population in vivo using a method such as Isl1-Cre/Wt1-Dre in a double reporter background (Rosa-Ai66) would be required to adequately evaluate the fate of this population in vivo. This is especially true since the majority of the paper evaluates the propensity of iPS derived Isl1+ CM to transform into adipocytes directly. Indeed, better characterization of the WT1-CreERT2;Rosa-mTmG mouse upon dosing with TMX at E7.5 (very early) is warranted. The WT1-CreGFP line is reported to have an early and transient ubiquitous expression domain just prior to E9.5. Since the Wt1-CreERT2 line is at the identical locus, one might worry that an early ectopic pattern may be present in this line as well, coincident with E7.5 TMX dose and leading to ectopic labeling.

After the first report by William Pu and colleagues in Nature 2008 that some Wt1-derived epicardial cells differentiated into cardiomyocytes during normal heart development, the contribution of WT1 lineage to CMs is still debated and we agree with the referee on that. Therefore, as suggested, we have performed a detailed characterization of the E8 Isl1/Wt1 lineage descendants (amount of cells and their specific localization) in the developing heart and proepicardial organ at E9.5 as well as in the myocardium and cardiac fat depot of adult mice, using Isl1^{MerCreMer/+};R26^{mTmG/+} and Wt1^{CreERT2/+};R26^{mTmG/+} animals upon dosing with tamoxifen at E7.5. These new extended analyses clearly demonstrate the existence, at both developmental time points, of a cellular overlap between the two lineage tracings in terms of cell types and their organ localization, strongly suggesting that these specific cells derive from a common progenitor that express both Isl1 and Wt1 at the time of tamoxifen injection. Moreover, we found that labelling of Wt1^{CreERT2/+};R26^{mTmG/+} E9.5 embryos was restricted to the proepicardial organ/epicardium and the urogenital tract, both Wt1-lineage derivatives, indicating the specificity of our marking by E7.5 tamoxifen injection. All these new data

are now described in the Result section on page 7, first paragraph, and illustrated in the new Expanded View Figures EV2 and EV3.

Since, to our knowledge, no Dre reporter lines have been yet generated for either the *Isl1* and/or the *Wt1* lineage, the lineage tracing with a combined Cre/Dre;*Rosa-Ai66* background suggested by the referee is practicably very time-consuming and exceeds the limited revision time. We agree that this approach would be the ideal one for quantifying the exact contribution of the *Isl1*⁺/*Wt1*⁺ lineage to the various heart structures *in vivo*. However, we believe that our new extended analyses of the E9.5 embryos and adult hearts of *Isl1*^{MerCreMer/+};*R26*^{mTmG/+} and *Wt1*^{CreERT2/+};*R26*^{mTmG/+} mice, in combination with our previous *in vitro* clonal experiments using mouse ESC-derived *Isl1*⁺/*Wt1*⁺ progenitors, provide very strong evidence that a cardiac precursor population expressing *Isl1* and *Wt1* exist, which is capable of generating both cardiomyocytes and adipocytes *in vivo* and *in vitro*.

2. Using iPS derived CM to suggest a novel cellular source of adipogenesis in ARVC is artifact prone. This immature CM population might be susceptible to differentiate into other cell types in vitro. Furthermore, ARVC is thought to become evident primarily in the adult following stress such as exercise. iPS-derived CM is not an adequate model of mature adult CM undergoing mechanical stress. Convincing evidence that CM contribute directly to the adipocyte population in vivo is lacking. It is difficult to imagine mature adult CM transdifferentiating into adipocytes in ARVC.

Previous fate mapping studies from the Ali Marian group have provided first evidence in mouse that mature adult cardiomyocytes can convert into adipocytes *in vivo* in the setting of human autosomal dominant ARVC, e.g. in presence of heterozygous deficiency of the desmosomal protein desmoplakin (Lombardi et al., *Circ Res* 2009). Using *aMHC*^{Cre/+};*Dsp*^{flx/flx};*R26*^{EYFP/+} mice, which allow simultaneous ablation of desmoplakin in adult cardiomyocytes and their lineage tracing, the authors detected few adipocytes labelled by the EYFP reporter. Interestingly, when desmoplakin was ablated earlier at the stage of cardiac progenitors using either *Mef2C*^{Cre/+};*Dsp*^{flx/flx};*R26*^{EYFP/+} or *Nkx2.5*^{Cre/+};*Dsp*^{flx/flx};*R26*^{EYFP/+} mice, not only the overall number of pathological “excess of adipocytes” was increased but also the percentage of adipocytes expressing EYFP. Since in the latter mice cardiomyocytes are irreversibly marked by EYFP, this could suggest a higher efficiency of cardiomyocyte-to-adipocyte conversion when ARVC-causing mutations “act” early during cardiogenesis. This is in agreement with our findings in patient iPSC-derived cardiomyocytes showing that “priming” for the cardiomyocyte-adipocyte switch occurs and this is likely at the level of a progenitor cell. We have now highlighted the study of Lombardi et al. in the Introduction section of the revised manuscript on page on page 3, second paragraph.

To further validate the capability of mature cardiomyocytes to transdifferentiate into adipocytes *in vivo* under the conditions experienced by ARVC myocytes (high level of PPAR α and nuclear WT1), we overexpressed NLS-WT1 and PPAR α (alone and in combination) in adult myocytes of *Myh6*^{Cre/+};*R26*^{mTmG/+} mice using adeno-associated viruses serotype 9 (AAV9-NLS-WT1 and AAV9-PPAR α), which display high tropism to cardiac and skeletal muscles. Tail vein injection of 2.5 x 10¹² virus particles resulted in 30-40 infected mG⁺ cells per heart slice. Interestingly, we observed emergence of few intramyocardial lipid-filled/*Plin*⁺ adipocytes labelled by mG exclusively in hearts of animals that received both viruses (2-3 *Plin*⁺/mG⁺ cells per heart slice at 5 weeks post-infection). Since WT1, differently from human, is not expressed in mouse cardiomyocytes, these results hint to a requirement of WT1 for the conversion process. This is also confirmed by our additional new data demonstrating that silencing of WT1 in human iPSC-CMs hampers their PPAR α -mediated adipocyte conversion (revised Figure 4, panels F and H, and new Appendix Figure S6). Moreover, they may explain the relative low amount of cardiac fat in mouse compared to human and the fact that all reported ARVC mouse models lacking specific desmosomal genes, although they develop arrhythmias, show only very limited adipogenic phenotypes.

These new data are now described in the Result section on page 10, second paragraph, and page 11, first paragraph, and illustrated in the new Expanded View Figure EV5 of the revised manuscript.

3. The population of cells supposed to be involved in vivo (Isl1 progenitors that no longer express Isl1 but do express Wt1) are not the same as the cells used in all in vitro experiments (cells that express both Isl1 and Wt1). The correlation between in vivo experiments and in vitro studies is therefore not strong. WT1 staining in control and ARVC patient hearts is not convincing and does not agree with the literature.

We apologize, but we do not clearly understand these specific points.

Considering that tamoxifen injection was performed at E7.5, we mark predominantly cells between E7.5 and E8.5, time-window preceding the PEO state and in which the Wt1+ cells still express Isl1, thus “counterparting” well the cells used in the in vitro experiments.

To the best of our knowledge, expression of WT1 in human hearts has been so far evaluated in aborted foetuses and discordant results have been reported. Parenti et al. (Acta Histochemica 2013) describe a strong and diffuse cytoplasmic staining for WT1 in cardiac muscle cells throughout the gestational period analysed (weeks 7 to 24). Also Ambu et al. (Eur J Histochem 2015) report a cytoplasmic immunoreactivity in the cardiomyocytes of all four chambers, with the atria muscle cells showing the strongest signal; no staining in the wall of the large arteries and endocardium was observed; nuclear expression in the epicardium was noticed in hearts from gestational age 7-9 weeks, while was absent in hearts from 11-12 weeks-old foetuses. In contrast, Duim et al. (J Mol Cell Cardiol 2016) show nuclear WT1 expression in epicardial, endothelial and endocardial cells varying in a spatiotemporal manner from week 4 till week 20; cardiomyocytes are reported to be negative.

Our immunocytochemistry analysis on hiPSC-derived CMs indicates expression of WT1 in both wt and PKP2- or MYH10-mutated cells, which was predominantly localized in the cytoplasm during culture in myogenic medium. This is in agreement with the results of Parenti et al. and Ambu et al. One study by Braitsch et al. (J Mol Cell Cardiol 2013) has evaluated the expression of several embryonic epicardial progenitor markers, including WT1, in the adult human heart of diseased donors with myocardial infarction, hypertension and congestive heart failure. However, the analysis was restricted only to the epicardial, perivascular and interstitial fibrotic tissue and no images of cardiac muscles cells are shown. Interestingly, WT1 expression was both nuclear and cytosolic in the fibrotic cells. We are not aware of any published data on WT1 expression in ARVC patient hearts and, in general, in the human myocardium of adult individuals. Our immunohistochemical analysis revealed a cytosolic expression of WT1 in CMs of non-ARVC patients, while expression was nuclear in some of the CMs adjacent to the fibrofatty tissue in ARVC hearts, corroborating the results on the hiPSC system.

4. Page 9 states that they see WT1 expression in CMs of ARVC patients. However this contradicts both lineage tracing and immunostaining studies, and also contradicts their reference (Ambu et al., 2015) that states WT1 is observed in fetal CM (not adult).

We would like to point out that study by Ambu et al. (please see also our response to point 3. above) was restricted only to the human foetal tissues and human adult heart was not analysed. To the best of our knowledge, there are no published data available on the expression of WT1 in human adult cardiomyocytes.

In mouse, the expression of Wt1 in the heart is well documented by several studies. The first cardiac expression of Wt1 protein is in the PEO, followed by the epicardium; Wt1 expression is detected in both embryonic and reactivated adult epicardium as well as in cardiac endothelial cells during development and after injury, but is absent in CMs at any developmental and adult stage.

Accordingly, our immunostaining analysis of Wt1 in mouse embryos at E9.5 revealed expression in the PEO, and no signal was detected in the CM of the developing heart (Figure 2B,c'). Expression of Wt1 was also identified in a rare population of Isl1 positive cells in the caudal lateral mesoderm at E8 (Figure 2C,iii). Moreover, our lineage tracing studies in Wt1^{CreERT2/+};R26^{mTmG/+} mice upon tamoxifen injection at E7.5 showed expression of the lineage marker mG in the PEO and epicardium at E9.5 (new Expanded View Figure EV2), while PEO derivatives (AV groove adipocytes, cardiac smooth muscle cells, cardiac fibroblasts, and epicardium) were marked in postnatal hearts at P28 (Figure 2D, Figure EV3, and Appendix Figure S2A), as expected. In 6 out of 7 P28 hearts, we also found mG⁺ CMs (Figure 2D and new Expanded View Figure EV3), supporting a previous study by Zhou et al. (Nature 2008) indicating that some myocytes of the heart originate from Wt1⁺ epicardial cells.

All together, considering that cardiac WT1 expression might be different between human and mouse (as the published data on the developing heart point out), we do not feel that our results in ARVC patients contradict the immunostaining and lineage tracing studies in the mouse.

5. The authors interpret reduced PKP2 levels observed as "compromised cell-cell adhesion" throughout the manuscript without evidence this is the case. Evaluating physical cell associations would be required. Loss of cellular contacts typically leads to MRTF accumulation in the nucleus, a contradictory finding that is not commented on. How does MRTF localization and PPARg expression respond to physical or chemical disruption of cell-cell contacts?

In response, we would like to point out that previous studies addressing a connection between MRTF/SRF and cellular contacts were performed in epithelial cells and mainly focused on E-Cadherin-based adherens junctions (Busche et al., J Cell Sci 2008 and 2010). These publications provide evidence for an activation of MRTF-A which is specifically mediated by E-Cadherin and a Rac1 (not RhoA)-driven depletion of actin monomers.

In contrast, our work provides first evidence for an (indeed contrary) impact of desmosomal junctions in human cardiomyocytes. Unlike upon disruption of epithelial adherens junctions (which results in upregulation of Rac1 and RhoA activity), we observe reduced levels of active RhoA in PKP2- and MYH10-mutated CMs characterized by desmosomal instability (Figures 1A and 7E, Figure EV1C, and Appendix Figure S8J). Accordingly, there seems to be a clear distinction between how epithelial adherens junctions and desmosomes of CMs connect to the activity state of small GTPases. Consistent with attenuated Rho-ROCK signalling in the mutated CMs, we further provide evidence for an increased concentration of monomeric actin (Figures 6C and 7F), which is fully in line with our observation of less nuclear MRTF-A and reduced MRTF/SRF transcriptional activity under these conditions.

We agree with the referee that for now, we can only speculate about the overall physical properties of contacts formed between mutated CMs. However, given the clearly changed appearance of desmosomal structures, we feel confident to at least reason a desmosomal instability with likely implications on the overall strength of intercellular contacts. It will be interesting to investigate whether acute loss of cellular contacts in our cells would mimic the effects of constitutive disrupted desmosomes in regards to MRTF and PPARg expression/localization. However, differently from adult CMs, physical or chemical disruption of cell-cell contacts in iPSC-derived CMs, as obtained by enzymatic single cell dissociation, leads to a dramatic disarray of the myofibrils and rounding up of the cells, thus rendering difficult to discriminate changes in subcellular localization of proteins.

Minor points:

6. Quantification of the % / number of WT1 / ISL1 positive cells, both by lineage tracing (Fig. 2) and by immunofluorescence (Fig. 3F) is required.

We have now specified the percentage of Wt1+ cells (in the PEO of $Isl1^{Cre/+};R26^{mTmG/+}$ embryos at E9.5) which also express the lineage marker mG in the Result section on page 7, first paragraph. The number of mESC-derived YFP+/Wt1+ cell clones is also indicated on page 8, first paragraph.

2nd Editorial Decision

26 March 2018

Thank you for submitting a revised version of your manuscript. The manuscript has now been seen by the three original referees, who find that their main concerns have been addressed and are now broadly in favour of publication of the manuscript. There remain only a few minor issues that have to be dealt with before I can extend formal acceptance of the manuscript:

1. Please address the remaining minor comment from referee #1.

REFeree COMMENTS

Referee #1:

I find this to be an impressive paper. The studies are rigorous and the results are very interesting. The authors have done a commendable job in addressing the reviewers concerns. My only minor

comment is that the Seahorse (respiration) studies in Fig. 11 are difficult to interpret. It seems that the CM cultures (ie. WT) have a very high proportion of uncoupled respiration (oligomycin-insensitive). Also, the respiratory activity of brown adipocytes cannot be adequately studied under basal conditions, without the addition of a β -agonist (or fatty acids) to induce UCP1-activity. My recommendation would be to remove this panel as it does not affect the overall conclusions of the paper. The morphological and molecular analysis of the cultures suffice here. Also, the wt and mut cultures are drastically different and thus comparisons between them in this assay are not very informative (comparing CM with brown fat-like cells). The paper should be published without further delay. Congratulations on a really elegant study.

Referee #2:

This manuscript by Dorn and colleagues has been significantly improved in revision and includes extensive new experimental data to strengthen the conclusions. Most notably the authors have further validated (by gene expression and functional parameters) the brown identity of the converted ARVC cardiomyocytes (CMs) and compared them with primary sources of brown/beige adipocytes. They have carried out a more thorough analyses of the *Isl1/Wt1* lineage in the developing heart and provide direct evidence that mouse adult CMs can convert into adipocytes *in vivo* under the influence of forced expression of PPAR γ and WT1.

With respect to my specific comments regarding more detail on the role of WT1 in the CM-adipocyte conversion, the authors have now demonstrated that shRNA targeting of WT1 in PPAR γ over-expressing CMs significantly reduced the incidence of adipocyte-like cells (Fig. 4F, H). In addition, they also reveal a direct physical interaction between WT1 and PPAR γ by co-immunoprecipitation studies (Fig. 4I). Collectively these data are important in implicating WT1 more directly in the process but also in demonstrating a cooperative role for WT1 and PPAR γ in driving CM-adipocyte conversion. Finally, they also include data (Appendix Fig S4) suggesting that mESC-derived *Isl1+/Wt1+* clones are more prone to adipogenesis than those lacking *Wt1*, which is highly informative regarding the potential origin of those CMs most susceptible to conversion in ARVC.

In summary, it remains the case that this study provides important insights into: i) the developmental origins of cardiac adipose tissue and a subset of cardiomyocytes; ii) the maintenance of adipose versus cardiomyocyte cell identity by mechano-sensing and iii) molecular mechanisms of cardiomyocyte-adipose conversion in ARVC through cooperative interaction of WT1 and PPAR γ .

Referee #3:

All initial comments have been adequately addressed.

2nd Revision - authors' response

29 March 2018

We have addressed all editorial comments and the comment from referee #1 outlined in the letter.

Referee 1.

I find this to be an impressive paper. The studies are rigorous and the results are very interesting. The authors have done a commendable job in addressing the reviewers concerns. My only minor comment is that the Seahorse (respiration) studies in Fig. 11 are difficult to interpret. It seems that the CM cultures (ie. WT) have a very high proportion of uncoupled respiration (oligomycin-insensitive). Also, the respiratory activity of brown adipocytes cannot be adequately studied under basal conditions, without the addition of a β -agonist (or fatty acids) to induce UCP1-activity. My recommendation would be to remove this panel as it does not affect the overall conclusions of the paper. The morphological and molecular analysis of the cultures

suffice here. Also, the wt and mut cultures are drastically different and thus comparisons between them in this assay are not very informative (comparing CM with brown fat-like cells). The paper should be published without further delay. Congratulations on a really elegant study.

We agree with the referee that the comparison of respiratory activity of CMs and brown fat-like cells is difficult to interpret. Therefore, we followed the referee's suggestion and removed the panel I of the Figure 1. This resulted in a modified Figure 1.

Corresponding Author Names: Alessandra Moretti and Karl-Ludwig Laugwitz

Manuscript Number: EMBOJ-2017- 98133